# Improving Results of Existing Groundwater Numerical Models Using Machine Learning Techniques: A Review

Cristina Di Salvo

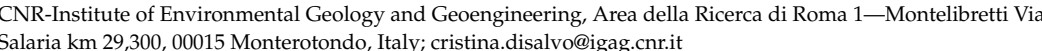

CNR-Institute of Environmental Geology and Geoengineering, Area della Ricerca di Roma 1—Montelibretti Via Salaria km 29,300, 00015 Monterotondo, Italy; cristina.disalvo@igag.cnr.it

**Abstract:** This paper presents a review of papers specifically focused on the use of both numerical and machine learning methods for groundwater level modelling. In the reviewed papers, machine learning models (also called data-driven models) are used to improve the prediction or speed process of existing numerical modelling. When long runtimes inhibit the use of numerical models, machine learning models can be a valid alternative, capable of reducing the time for model development and calibration without sacrificing accuracy of detail in groundwater level forecasting. The results of this review highlight that machine learning models do not offer a complete representation of the physical system, such as flux estimates or total water balance and, thus, cannot be used to substitute numerical models in large study areas; however, they are affordable tools to improve predictions at specific observation wells. Numerical and machine learning models can be successfully used as complementary to each other as a powerful groundwater management tool. The machine learning techniques can be used to improve calibration of numerical models, whereas results of numerical models allow us to understand the physical system and select proper input variables for machine learning models. Machine learning models can be integrated in decision-making processes when rapid and effective solutions for groundwater management need to be considered. Finally, machine learning models are computationally efficient tools to correct head error prediction of numerical models.

**Keywords:** groundwater; physically-based models; machine learning models; artificial neural network; random forest; support vector machine





## 1. Introduction

### 1.1. Physically Based Models in Groundwater Management

Physically-based models are the most commonly used tools in quantitative groundwater flow and solute transport analysis and management. Traditionally, the conceptual or numerical models are applied to hydrological modelling in order to understand the physical processes characterising a particular system, or to develop predictive tools for detecting proper solutions to water distribution, landscape management, surface water–groundwater interaction, or impact of new groundwater withdrawals. Along with the ever rising accessibility of computational power, field measurements, and improved understanding of the dynamics of hydrogeological systems, the accuracy required for these models is increasing. This brings some practical limitations of physically-based based models, including the need for large amount of data and input parameters [1,2]. In order to solve the equations describing the dynamics of flow, the physical properties as well as the boundary conditions of the system must be suitably defined within the time and space domains of the model in order to achieve acceptable accuracy. Quantifying these properties and conditions can be expensive and time-consuming; thus, very few field measurements are often available, and the accurate estimate of model parameters across the study area can be challenging [3].

### 1.2. Uncertainty and Error Types of Physically-Based Models

Many modellers recognised the inherent uncertainty of physically-based models (e.g., refs. [4–6]. They are subject to three types of errors: (1) model structural error introduced by misrepresentation of the real system, as well as from the numerical implementation, for example, spatial and temporal discretisation [4,7,8]; (2) parameter error due to indirect estimation (e.g., prior knowledge or calibration) [9,10]; (3) errors in input data [11] and measurements used to evaluate the model. Alternatively, when the target is to obtain accurate predictions rather than understanding the underlying groundwater system, conventional statistical techniques, such as autoregressive (AR), AR moving average (ARMA), and AR integrated moving average (ARIMA) have been applied invariably to modelling groundwater resources [12,13]. However, the abovementioned methods do not take into account the nonstationary and non-linear characteristics of the data structure [14,15].

### 1.3. Machine Learning Models

The need to address groundwater problems through alternative, relatively simpler modelling techniques pushed authors in different parts of the world to explore machine learning models. Machine learning methods have been widely used in recent years in many fields (i.e., bioinformatics, biomedicine [16,17], biochemical engineering [18], civil engineering problems, see refs. [19–21] and references therein), transportation networks [22–24], geosciences and environmental applications [25–27], and environmental risk prediction [28,29]. Their largely diffused uses are due to the fact that they are simple and provide acceptable results. Recently, the modelling of non-linear and non-stationary problems has been provided with great ability by machine learning techniques compared with traditional statistical approaches [30–34]. Dealing with machine learning techniques, modellers do not need to introduce the mathematical relationships among variables because machine they are capable of learning the relationships from the input data. Of course, these methods have some limitations, such as overtraining leading to low generalisability [35], risk of using unrelated data, incorrect modelling with inappropriate methods, their dependency on data for training [36], and so on. However, their simplicity of use, high-speed run and reasonable accuracy without the need to know the physics of the problem have led many researchers to apply them.

### 1.4. Machine Learning for Groundwater Level Forecasting: Current State of the Research

Many recently published review papers have explored the use of machine learning models in hydrology (e.g., refs. [37–41] and references therein, refs. [42,43], or in many water resources fields (e.g., refs. [44–47] specifically for groundwater level (GWL) modelling and forecasting, refs. [48,49] and references therein)). However, there is not yet a complete review paper examining the application of machine learning methods in GWL modelling in comparison to numerical models. The development of better approaches for GWL modelling makes it necessary to look at what has been done in the field of the comparison of numerical and machine learning models and current research.

### 1.5. Aim of This Work

This paper presents a review of those papers specifically focused on the use of both numerical and machine learning methods for groundwater modelling to estimate the groundwater levels. The aim of the paper was to furnish information to orient modellers which want to explore machine learning approaches starting from an already developed numerical model, highlighting the advantages and disadvantages of both modelling techniques. Moreover, it attempts to clarify some common questions such as: which machine learning techniques are appropriated to solve a specific problem; which is the optimal input data range for machine learning modelling; and which software is suitable for a specific machine learning model. In the following chapters, the types of physically based models used in the reviewed papers are briefly described. Then, some commonly used machine learning methods for modelling GWL are addressed. The methods include Arti-

ficial Neural Networks, Radial Basis Function, Adaptive-Neuro Fuzzy Inference System, Time Lagged Recurrent neural networks, Extreme Learning Machine, Bayesian Network, Instance-Based Weighting, Inverse-Distance Weighting, Support Vector Machine, Decision Tree, Random Forest, Gradient-Boosted Regression Tree, and some hybrid models such as wavelet-machine learning models. Radial Basis Function, Adaptive-Neuro Fuzzy Inference System, Time Lagged Recurrent neural networks and Extreme Learning Machine, Bayesian Network, Instance-Based Weighting, Inverse-Distance Weighting, Support types of Artificial Neural Networks, Random Forest, and Gradient-Boosted Regression Tree are types of Decision Trees; however, in this review, each technique was treated individually. The most frequently used machine learning techniques used are Artificial Neural Networks, Bayesian Network, Decision Tree, and Support Vector Machine. At first, each method is briefly described and thereafter the related studies are reviewed. This is followed by general and specific results, discussions, and conclusions, including recommendations for future research.

## 2. Modelling Techniques Explored in This Review

### 2.1. Physically Based Numerical Groundwater Flow Models

Numerical groundwater flow models simulate the distribution of head by solving the equations of conservation of mass and momentum. Because these equations represent the physical flow system, in order to obtain accurate results accuracy, the physical properties of the aquifer (e.g., hydraulic conductivity, specific storage) as well as the initial and boundary conditions of the system must be properly assigned within the time and space domains of the model [3]. The physically based models used in the reviewed papers are briefly described as follows.

MODFLOW [50,51] is the modular finite difference flow model distributed by the U.S. Geological Survey. It is one of the most popular groundwater modelling programs. Thanks to its modular structure, MODFLOW integrates many modelling capabilities to simulate most types of groundwater modelling problems. The corresponding packages (e.g., solute transport, coupled groundwater/surface-water systems, variable-density flow, aquifer-system compaction and land subsidence, parameter estimation) are well structured and documented and can be activated and used to solve required modelling problems. The source code is free and open source, and can be fixed and modified by anyone with the necessary mathematical and programming skills to improve its capabilities [52].

SUTRA (Saturated-Unsaturated Transport) [53] is a 3D groundwater model that simulates solute transport (i.e., salt water) or temperature. The model employs a grid that is based on a finite element and integrated finite difference hybrid method framework. The program then computes groundwater flow using Darcy's law equation, and solute or transport modelling use similar equations. It is very frequently used for calculation of salinity of infinite homogeneous, isotropic unconfined aquifer.

The Princeton Transport Code (PTC, [54,55] is a 3D groundwater flow and contaminant transport simulator. It uses a hybrid coupling of the finite-element and finite-difference methods. The domain is discretised by the algorithm into parallel horizontal layers; the elements within each layer are discretised by finite-element method. The vertical connection between layers is allowed by a finite-difference discretisation. During any iteration, all the horizontal finite-element discretisations are firstly solved independently of each other; then, the algorithm solves the vertical equations connecting the layers using the solution of the horizontal equation.

SHETRAN is a physically-based distributed modelling system for simulating water flow, sediment, and contaminant transport in river basins [56]. It is often used to model integrated groundwater–surface water systems. SHETRAN simulates surface flows using a diffusive wave approximation to the Saint–Venant equations for 2D overland flow and 1D flow through channel networks. Subsurface flows are modelled using a 3D extended Richards equation formulation, where the saturated and unsaturated

zones are represented as a continuum. Surface and subsurface flows exchange is allowed in either direction. The partial differential equations for flow and transport are solved on a rectangular grid by the finite difference methods; the soil zone and aquifer are represented by cells which extend downwards from each of the surface grid elements. Precise river–aquifer exchange flows can be represented by using the local mesh refinement option near river channels.

### 2.2. Machine Learning Models

#### 2.2.1. Artificial Neural Networks (ANNs)

An artificial neural network (ANN) model is a data-driven model that simulates the actions of biological neural networks in the human brain. Typically, an ANN comprises a variable number of elements, called neurons, which are linked by connections. Generally, an ANN is composed of three separate layers: input, hidden, and output layers. Each single layer contains neurons with similar properties. The input layer takes input variables (e.g., past GWL, temperature, precipitation time series); a relative weight (i.e., an adaptive coefficient) is given to each input, which modifies the impact of that input. In the hidden and output layers, each neuron sums its input, and then applies a specific transfer (activation) function to calculate its output. By processing historical time series, the ANN learns the behaviour of the system. An ANN learns by relating a given number of input data with a resulting set of outputs [57], which is the training process. Training means modifying the network architecture to optimise the network performance, which involves tuning the adjustable parameters: tuning the weights of the connections among nodes, pruning or creating new connections, and/or modifying the firing rules of the single neurons [58]. The training process can be conducted with various training (learning) algorithms. ANN learning is iterative, comparable to the human learning from experience [59]. ANNs are very popular for hydrologic modelling and is used to solve many scientific and engineering problems. These models may be ascribed to two categories: feed-forward, which is the most common, and feed-back networks [60,61]. The most frequently used family of feed-forward networks is the multilayer perceptron [62,63]; it contains a network of layers with unidirectional connections between the layers.

#### 2.2.2. Radial Basis Function Network (RBF)

RBF network is commonly a three-layer ANN which uses RBF as activation functions in the hidden layer; the network architecture is the same as multilayer perceptron. The number of neurons in the input layer is the same as the input vectors. The radial basis functions in the hidden layer map the input vectors into a high-dimension space [64]. A linear combination of the hidden layer outputs is used to calculate the neurons in the output layer of the network. The distinctive characteristic of RBF is that the responses increase (or decrease) monotonically with Euclidean distance between the centre and the input vectors [65].

#### 2.2.3. Adaptive Neuro-Fuzzy Inference System (ANFIS)

ANFIS, first described by Jang [36], combines the neural networks with the fuzzy rule-based system. In the fuzzy systems, relationships are represented explicitly in the form of if-then rules [66,67]. Different from a typical ANN, which uses sigmoid function to convert the values of variables into normalises values, an ANFIS network converts numeric values into fuzzy values. Firstly, a fuzzy model is developed, where input variables are derived from the fuzzy rules. Then, the neural network tweaks these rules and generates the final ANFIS model [68]. Usually, an ANFIS model is structured by five layers named according to their operative function, such as 'input nodes', 'rule nodes', 'average nodes', consequent nodes', and 'output nodes', respectively [69].

### 2.2.4. Time Lagged Recurrent Neural Networks (TLRNs)

TLRN are multilayer perceptrons extended with "short-term" memory structures that have local recurrent connections. The approach in TLRNs differs from a regular ANN approach in that the temporal nature of the data is taken into account [69], allowing accurate processing of temporal (time-varying) information. The most common structure of a TLRN comprises an added feedback loop which introduces the short-term memory in the network [70] so that it can learn temporal variations from the dataset [71]. TLRN uses a more advanced training algorithm (back propagation through time) than standard multilayer perceptron [72]. The main advantage is that the network size of TLRNs is lower than multilayer perceptrons that use extra inputs to represent the past state of the system. Furthermore, TLRNs have a low sensitivity to noise.

### 2.2.5. Extreme Learning Machine (ELM)

ELM is a training algorithm for the single-layer feed-forward-neural network (SLFFNN). Input weights and biases values of the nodes in the hidden layer are randomly determined according to continuous probability distribution with probability of 1, so as to be able to train N separate samples. Compared with conventional neural networks, in ELM, only the number of hidden layer neurons needs to be tuned, and no adjustments are required for parameters such as learning rate and learning epochs. Training of ELM is conducted quickly and is considered a universal approximator [73–75].

### 2.2.6. Bayesian Network (BN)

The Bayesian networks (Figure 1) are statistical-based models which compute the conditional probability associated with the occurrence of an event by using the Bayes' rule. A typical Bayesian network is composed of a set of variables where their conditional dependencies are represented by a directed acyclic graph.

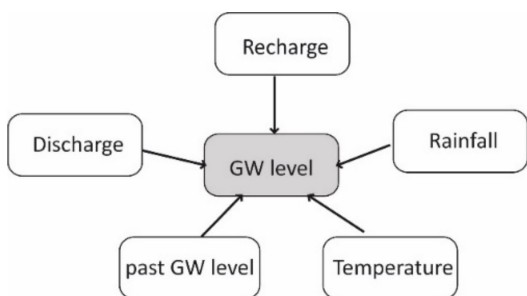

**Figure 1.** Example of the structure of a Bayesian model applied to groundwater-level study.

Connections define the conditional dependencies among variables (i.e., nodes) [76]. The dependencies are quantified by conditional probabilities for each node through a conditional table of probabilities. Usually, BNs are built by software that generates many network structures with the input parameters.

### 2.2.7. Instance-Based Weighting (IBW)

Instance-based algorithms derive from the nearest-neighbour pattern classifier [77], which is modified and extended by introducing a weighting function. IBW models are also inspired by exemplar-based models of categorisation [78]. Different from other machine learning algorithms, which return an explicit target function after learning from the training dataset, instance-based algorithms simply save the training dataset in memory [79]. For any new data, the algorithm first finds its n nearest neighbour in the training set and delays the processing effort until a new instance needs to be classified. IBW has many advantages such as the low training cost, the efficiency gained through solution reuse [80], ability to model complex target functions, and the capability to describe probabilistic concepts [81]. However, when irrelevant features

are present, their performance decreases; an accurate distinction of relevant features can be achieved through feature weighting to ensure acceptable performance. IBW does not need to be trained and the results are less influenced by the training data size. Inverse-distance weighting is a special case of instance-based weighting with the weighting factor p = 2 [82].

### 2.2.8. Support Vector Machine (SVM)

SVM are kernel-based neural networks developed by Vapnik [83] to overcome the several weaknesses which affect the ANNs' overall generalisation capability [84], including possibilities of getting trapped in local minima during training, overfitting the training data, and subjectivity in the choice of model architecture [85]. The SVM is based on statistical learning theory [86]; in particular, it is based on structural risk minimisation (SRM) instead of empirical risk minimisation (ERM) of ANNs. The SVM minimises the empirical error and model complexity simultaneously, which can improve the generalisation ability of the SVM for classification or regression problems in many disciplines. This is achieved by minimising an upper bound of the testing error rather than minimising the training error [79]; the solution of SVM with a well-defined kernel is always globally optimal, while many other machine learning tools (e.g., ANNs) are subjected to local optima; finally, the solution is represented sparsely by Supporting Vectors, which are typically a small subset of all training examples [87]. For further details, see refs. [63,86,88,89].

### 2.2.9. Decision Trees (DT)

Decision tree models [90] are based on the recursive division of the response data into many parts along any of the predictor variables in order to minimise the residual sum of squares (RSS) of the data within the resulting subgroups (i.e., "nodes" in the terminology of tree models) [91]. The number of nodes increases during the process of splitting along predictors. The tree-growing process stops when the within-node RSS is below a specified threshold or when a minimum specified number of observations within a node is reached [92]. However, the modeller places minimal limitations upon tree-fitting process, and fitted trees may be more complex than is actually warranted by the data available. The problem of overfitting results is then managed by the 'pruning' algorithms, which aid the modeller in the selection of a parsimonious description of interactions between response and predictors, fitting trees for the optimum structure for any level of complexity [91]. Because no prior assumptions are made about the nature of the relationships among predictors, and between predictors and response, decision trees are extremely flexible.

### 2.2.10. Random Forest (RF)

Random forests work by constructing groups of decision trees during the training process, representing a distinct instance of the classification of data input. Each tree is developed by independently sampling the values of a random vector with the same distribution for all trees in the forest [93].

The random forest technique considers the instances individually so that the trees are run in parallel; there is no interaction between these trees while building the trees. The prediction with the majority of votes or an average of the prediction is taken as the selected prediction (Figure 2). The RF algorithm was created to overcome the limitations of DT, reducing the overfitting of datasets and increasing prediction accuracy. The decision tree grows to the largest possible size without being pruned in accordance with the number of trees and the number of predictor variables [94].

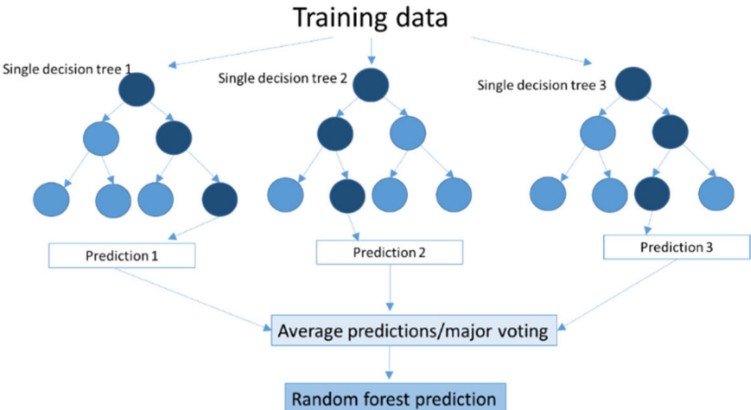

**Figure 2.** Scheme of Random Forest.

### 2.2.11. Gradient-Boosted Regression Trees (GBRT)

Gradient-Boosted Regression trees are ensemble techniques in which weak predictors are grouped together to enhance their performance [95]. Learning algorithms are combined in series to achieve a strong learner ("boosting") from different weak learners (i.e., the decision trees) connected sequentially. Each tree attempts to minimise the errors of the previous tree. After the initial tree is generated from the data, subsequent trees are generated using the residuals from the previous tree. At each step, trees are weighted, with the lower-performing trees weighted the highest; this allows the improvement of performance at each iteration. A variety of loss functions can be used to detect the residuals.

### 3. Bibliographic Review

The following section describes the reviewed papers. Throughout our research, few papers were found in the literature that examine the use of both numerical models and machine learning models in GWL forecasting. Here, 16 papers dealing with the use of both models for the prediction of GW levels, which were published in 10 international journals and 1 book from 2003 to 2020, were reviewed. Each paper was analysed in detail; for each one, the author provided a description of the study area and the geological context, the area of model use (e.g., groundwater planning and supply, management in farming systems, coastal water management), the machine learning technique, and any details of its application in the specific case study. Finally, the statistical indicators used to compare the performance of numerical and machine learning models were reported. In the reviewed papers, machine learning models are always used to improve the results of physically-based models in GWL forecasting and to overcome the problem of long computational time of regional models. This is accomplished by comparing results of a physically-based model and a surrogate machine learning model (i), comparing results of a physically-based model and different machine learning models (ii), testing hybrid or ensemble models (iii), and reducing and correcting physically-based model errors by means of machine learning approaches (iv). In the cases (i) and (ii), each model is run independently. In the case (iii), machine learning techniques are applied at different stages of the modelling procedure, such as data pre-processing; in some papers, numerical model output is used to train machine learning models, obtaining statistical models capable of speeding up the numerical model runs. In case (iv), numerical model errors are used as training datasets for machine learning models. Details of the selected papers are given in Table 1, which includes information such as the region of study, the key area of model use, the used machine learning model, the hydrologic input variables of machine learning models, the time step, the range of total data, the total simulation time, time step, and the grid size of the physically-based model (Journal Citation Reports, Clarivate Analytics). In some cases, lacking information was integrated with literature complementary to the reviewed papers (i.e., same study areas).

**Table 1.** List of the reviewed studies. P: precipitation. T: temperature. E: evapotranspiration. SWL: surface water level. EP: effective precipitation.

| n | Reference | Region of Study (Country) | Key Area of Model Use | Used ML Models | Input Variables to ML Model | ML Model Time Step | Range of Total Data (Number of Data or Observation Wells) | PB Simulation Time (Time Step) | Size of the PB Model Domain | Field of Application of the ML Technique | Journal (202+C:L0 IF) | Aquifer or Basin Hydros-tratigraphy |
|---|---|---|---|---|---|---|---|---|---|---|---|---|
| 1 | Mohammadi, 2009 [59] | Chamchamal plain (Iran) | Farming and agricoltural systems | ANN | MODFLOW output | monthly | 1986–1998 (144 sets) | 1 year (monthly) | 145.7 Km$^2$ | using the results of PB models to train a single ML model | Pratical hydroinfor-matics (book) | Alluvial (karst bedrock) |
| 2 | Coppola et al., 2003 [3] | Northwest Hillsborough Wellfield (USA) | Planning and supply | ANN | GWL, pumping rates, P, T, dew point, wind speed conditions, stress period lenghts | weekly | January 1995–August 2000, (212 sets) | 20 years (monthly) | 10,359.9 Km$^2$ | using the results of PB models to train a single ML model | Journal of hydrologic engineering (2.064) | Highly permeable limestone overlain by low permeability clay and, above, sand with interbedded clay |
| 3 | Banajeree et al., 2011 [96] | Kavaratti, island of the Lakshadweep archipelago (India) | Coastal water management | ANN | not mentioned in the paper | monthly | 2005–2007 (23 sets) | 5 years (monthly) | 2D model, with section lenght = 2650 m and depth 1000 m | using the results of PB models to train a single ML model | Journal of hydrology (5.722) | Coastal |
| 4 | Mohanty et al., 2013 [97] | Kathajodi-Surua Inter-basin of Odisha (India) | Coastal water management | ANN, TLRNs | GWL, P, E, river stage, SWL, pumping rates | weekly | February 2004–May 2007 (174 sets) | 3 years, (weekly) | 114.5 m$^2$ | using the results of PB models to train a single ML model | Journal of hydrology (5.722) | Alluvial |
| 5 | Parkin et al., 2007 [98] | Winterbourne stream, Thames Basin, Berkshire (UK) | aquifer-river interaction | ANN | GWL, river flow depletion | daily | Not specified (1 well) | 25 years (daily) | Regional aquifer: 200 Km$^2$. Valley aquifer: 2 Km$^2$ | using the results of PB models to train a single ML model | Journal of hydrology (5.722) | Alluvial |
| 6 | Moghaddam et al., 2019 [76] | BirjandAquifer, South Khorasan (Iran) | Drought-prone regions | ANN, BN | GWL, E, T, EP, Discharge | monthly | 2002–2014 (1872 sets) | 12 years | 277.8 Km$^2$ ([99]) | using the results of PB models to train and compare different ML models | Groundwater for sustainable develop-ment (no IF) | Alluvial |

Table 1. *Cont.*

| n | Reference | Region of Study (Country) | Key Area of Model Use | Used ML Models | Input Variables to ML Model | ML Model Time Step | Range of Total Data (Number of Data or Observation Wells) | PB Simulation Time (Time Step) | Size of the PB Model Domain | Field of Application of the ML Technique | Journal (202+C:L0 IF) | Aquifer or Basin Hydros-tratigraphy |
|---|---|---|---|---|---|---|---|---|---|---|---|---|
| 7 | Almuhaylan at al., 2020 [68] | Saq Aquifer in Quassim (Saudi Arabia) | Drought-prone regions | ANN, ANFIS | GWL, pumping rates | not specified | 1980–2018 (55 wells) | not specified | 600 Km$^2$ | using the results of PB models to train and compare different ML models | Water (3.103) | Sandstone |
| 8 | Chen et al., 2020 [63] | Heihe River Basin (China) | Drought-prone regions | ANN, RBF, SVM | pumping rates, recharge, streamflow rates | monthly | 1986–2008 (11,088 sets) | 22 years (monthly) | 21,120 Km$^2$ | using the results of PB models to train and compare different ML models | Scientific reports (4.380 | Alluvial |
| 9 | Fienen et al., 2016 [95] | Lake Michigan Basin (USA) | Planning and supply | ANN, GBRT, BN | parameters expected to have predictive power to the source of water to wells | not specified | 1864–2005, (4911 sets) | 141 years (variable, [100]) | 204,764.4 Km$^2$ | using the results of PB models to train and compare different ML models | Environmental modelling and software (5.288) | Glacial deposits |
| 10 | Miro et al., 2021 [101] | San Bernardino and Rialto-Colton basins, San Bernardino Valley Municipal Water District - Valley District (USA) | Drought-prone regions | RF, SVM, ANN | Recharge, pumping rates | not specified | 2015–2050 (not specified) | 35 years (monthly) | 3000 Km$^2$ | using the results of PB models to train and compare different ML models | Climate risk manage-ment (4.090) | Basin comprising ancient metamorphic bedrock, eolic sands, ancient fans, recent alluvium |
| 11 | Malekzadeh et al., 2019 [102] | Kabodarahang Plain, Hamadan (Iran) | Farming and agricoltural systems | ELM, WA-ELM | decomposed sub-series of observed GWL | monthly | August 1990–September 2015 (301 sets) | 10 years (monthly) | not specified | using the results of PB models to train a single ML model | Groundwater for sustainable develop-ment (no IF) | Alluvial (limestone bedrock) |

**Table 1.** *Cont.*

| n | Reference | Region of Study (Country) | Key Area of Model Use | Used ML Models | Input Variables to ML Model | ML Model Time Step | Range of Total Data (Number of Data or Observation Wells) | PB Simulation Time (Time Step) | Size of the PB Model Domain | Field of Application of the ML Technique | Journal (202+C:L0 IF) | Aquifer or Basin Hydros-tratigraphy |
|---|---|---|---|---|---|---|---|---|---|---|---|---|
| 12 | Nikolos et al., 2008 [103] | Northern Rhodes Island (Greece) | Coastal water management | ANN combined with DE algorithm | GWL, pumping rates | daily | 1997–1998 (3125 sets) | 1 year (2 seasonal stress periods) | 217 Km$^2$ | using the results of PB models to test hybrid or ensamble modelling approaches | Hydrological processes (3.565) | Coastal |
| 13 | Sahoo et al., 2017 [104] | High Plains aquifer and Mississippi River Valley aquifer (USA) | Farming and agricoltural systems | Automated hybrid artificial neural network (HANN) | GWL, P, T, streamflow, climate indexes, irrigation demand, NAO index | monthly | 1980–2012, (HPA: 263,808 sets. MRVA: 115,368 sets) | 33 years (monthly) | MRVA: 405,720 Km$^2$ ([105]). HPA: 3.34 Km$^2$ ([106]) | using the results of PB models to test hybrid or ensamble modelling approaches | Water resource research (5.240) | High Plain Aquifer: ancient alluvial fans and quaternary deposits. Mississippi River Valley Alluvial Aquifer: Tertiary and Quaternary clay, silt, sand and gravel deposits. |
| 14 | Michael et al., 2005 [82] | Argonne National Laboratory, Illinois (USA) | Contaminant/phytoremed-iation | DT, IDW, ANN | GWL, P | quartely | November 1999–March 2001 (22 wells with quarterly data); May 2001 (7 wells with hourly data) | 6 years, (monthly) | 4.8 Km$^2$ ([107]) | using the results of PB models to test hybrid or ensamble modelling approaches | Water resource research (5.240) | Glacial deposits |
| 15 | Xu et al., 2014 [79] | Republican River Basin and Spokane Valley-Rathdrum Prairie aquifer (USA) | Planning and supply | Cluster analysis, IBW, SVM | GWL, well location, observation time | monthly | RRCA: 1918–2007 (300,000 sets). SVRP: 1990–2005 (2191 sets) | RRCA: 89 years. SVRP: 15 years, (monthly) | RRCA: 79,396 Km$^2$. SVRP: 844.3 Km$^2$ | using ML techniques for PB models errors reduc-tion/correction | Groundwater (2.671) | Alluvial |
| 16 | Demissie et al., 2009 [85] | Argonne National Laboratory, Illinois (USA) | Contaminant/phytoremed-iation | ANN, DT, SVM, IBW | GWL, EVP, stress periods | monthly | 2000–2005, (3600 sets) [107] | 6 year (monthly) | 0.75 Km$^2$ | using ML techniques for PB models errors reduc-tion/correction | Journal of hydrology (5.722) | Glacial deposits |

### 3.1. Comparing Results of a Physically-Based Model and a Surrogate Machine Learning Model

Many authors compared results of physically-based models and machine learning models run independently. The two approaches are then compared in terms of GWL prediction performance.

Mohammadi et al. [59] investigated the applicability of ANN models in simulating GWL for aquifers with limited data. The study area was the Chamchamal plain (Iran), an alluvial plain surrounded by a karstic formation. Groundwater flow was simulated by MODFLOW and hundreds of data sets were generated from the calibrated model to train the ANN model. Another purpose was to detect ANN models capable of simulating the complex dynamics of GWL, even with relatively short lengths of training data of the ANN model. To achieve this objective, different ANN models were implemented, with different combinations of input data. Furthermore, different network architectures, with different number of hidden layers and activation functions, were evaluated. The models' performances were evaluated by means of MODFLOW outputs and measured groundwater levels through the coefficient of determination ($R^2$), mean squared error (MSE), and normalised mean squared error (NMSE). The water table was estimated with reasonable accuracy by all the models, but the ANN required lesser input data and took less time to run. However, the authors remarked two disadvantages of these networks: (i) the water table cannot be predicted in all observation wells by a single model with similar input parameters; and (ii) models are static and inputs and outputs from previous time steps are not considered (unless these are introduced explicitly). This results in a high difference between the observed and calculated GWL at some points. In order to overcome these difficulties, the authors tested TLRN to simulate the entire groundwater system with one model. The aim of TLRN is to predict a multivariate time series using past values and available covariates. Instead of using static feed-forward ANNs to model nonlinear relationships in water table level forecasting, the TLRNs approach takes into account the temporal nature of the data (i.e., the lagged inputs, see Section 2.2.4), and in this respect compares favourably with ANN multilayer perceptron networks. The model used in the TLRNs is the gamma model [71], which is characterised by a memory structure that is a cascade of leaky integrators. The neural network can control the depth of the memory by changing the value of the feedback parameter, instead of changing the number of inputs. Since the feedback parameter is recursive, a backpropagation through time algorithm was used to apply a more powerful learning rule. Considering the reduced computational costs and the lower data requirements, the authors concluded that a TRLN model can be effectively used in the field of GWL simulation.

In the work of Coppola et al. [3] ANNs are used to accurately forecast transient water levels in a complex groundwater system under variable aquifer stresses. The model was tested in the Northwest Hillsborough Wellfield near Tampa Bay, Florida, USA, the model area being represented by the Upper Floridian aquifer (consisting of high permeability karst limestone overlain by a low permeability semiconfining unit, with a surficial sandy unconfined aquifer above). Results of numerical and machine learning models were compared for representative monitoring wells by using root mean square error and absolute mean error. The oscillation of the water levels was modelled with much more accuracy by ANN than the numerical flow model. The Absolute Mean Error of numerical model exceeded the maximum ANN prediction error at any single observation during each stress period. The authors concluded that for certain problems, ANN represents a better option to numerical modelling approaches because it does not require difficult-to-quantify aquifer parameters and time- and space-variable conditions. Then, three types of sensitivity of ANN were evaluated: (1) the sensitivity of ANN prediction performance to training set size; (2) sensitivity analysis of selected ANN inputs on water level responses; (3) sensitivity of ANN performance to data noise and measurement error.

(1) The sensitivity of ANN performance to data availability was assessed by using different sizes of training sets. The results showed that, during validation, acceptable prediction accuracy was achieved with a relatively small number of training sets.

(2) Input parameters groundwater withdrawals and rainfall were included in the sensitivity analysis. Results showed that, in the unconfined aquifer, short-term oscillations were correlated most strongly to rainfall, while in the underlying semiconfined aquifer the water level was mostly influenced by withdrawals. Since these results are in accordance with the hydrological conditions, the authors concluded that the physical dynamics of the system must be sufficiently understood by the modeller in order to identify the important predictor input variables.

(3) The effect of measurement error and data noise (inherently present in most hydrologic data set) on ANN performance was assessed by introducing normally distributed random noise into the input variables of the training set. The results demonstrated that the ANN can filter out noise in the training data and effectively learn groundwater system behaviour.

Banerjee et al. [96] evaluated the use of ANN simulation over mathematical modelling as a management tool for coastal aquifers. The aim of the models was to forecast the increase in the salinity of groundwater due to pumping at different rates in the island of Kvaratti, Lakshadweep archipelago (India) and to detect management strategies to avoid the increase of salinity of groundwater. A physically-based 2D finite element model was developed with SUTRA [53]. The study demonstrated the superiority of ANN with respect to the physically-based model, evaluated by mean of root mean squared error (RMSE) and mean absolute error (MAE). Its non-linear nature makes it a formidable tool for analysing real-world data, allowing modelling of complex dependencies. With respect to traditional models such as SUTRA, ANN requires a lesser number of input parameters and avoids the model building and parameter estimation phases. While only a few seconds are needed for the training in the ANN models, modelling in SUTRA is very time-consuming.

Mohanty et al. [97] compared the results of the finite difference-based numerical MODFLOW model and the ANN model in simulating GWL in an alluvial aquifer system (Kathajodi-Surua Inter-basin of Odisha, India) for improving the efficiency of planning and management of groundwater resource at the basin scale. To evaluate the results, 6 statistical criteria were used: bias, coefficient of determination ($R^2$, MAE, RMSE), Nash–Sutcliffe efficiency (NSE), and mean percent deviation (Dv). Results revealed that the ANN model performed better for short-time predictions that require high accuracy, while numerical models were more appropriate for long-term predictions. Furthermore, the authors highlighted that physically based models provide the total water balance of the system, whereas the ANN models do not involve a description of the entire physics of the system. In the case of ANNs, a new model must be developed from the beginning to include any changes in the input or output parameters, differently from numerical models. Thus, the type of model should be selected in accordance with the type of problem.

Parkin et al. [104] developed and tested an approach in which numerical and ANN models were used to evaluate the impacts of groundwater withdrawals on river flows in areas representing the hydrogeologic settings of most of England and Wales. Several ANN hidden node structures were tested. The ANN model was trained using the input and output data from about 2000 simulations of the SHETRAN numerical modelling system. The outputs of ANN model were compared against analytical models, and tested using a field data from a case study site: the Winterbourne stream within the Thames Basin near Reading, Berkshire, flowing across a chalk fractured aquifer. The parameters used for the ANN model come from many sources and comprise the distance of borehole from river, the aquifer transmissivity and storage coefficient, the valley-fill transmissivity and specific yield, the river width, the hydraulic conductivity and thickness of riverbed, and the mean annual recharge and the date of peak recharge. The performance was evaluated by comparing root mean square errors of normalised outputs. The results showed the successful application of the approach for modelling river–aquifer interactions and its potential for modelling complex hydrological systems. The good correspondence between the simulated and observed flow depletion using independently-derived parameter values demonstrates that this approach can be applied for modelling realistic field conditions.

*3.2. Comparing Results of a Physically-Based Model and Different Machine Learning Models*

Even if ANNs are widely used among machine learning technique by groundwater modellers, their limitations encouraged authors to explore alternative models to achieve better performance in GWL prediction.

Moghaddam et al. [76] compared a MODFLOW, an ANN, and a BN model to determine the most accurate method for simulating GWL in the alluvial Birjand aquifer, located in an arid region of the eastern Iran, and solve the problem of GWL overestimation of the MODFLOW model. Both BN and ANN models provided a reliable prediction for GWL. The BN model showed the best match between the measured and the predicted groundwater level values, evaluated by comparing $R^2$, RMSE, and NASH, and the best performance evaluated by a 2-year period groundwater hydrograph. BN models showed many advantages, such as the easier implementation, the higher forecasting accuracy, and the ability to deal with missing or incomplete data. Moreover, in the BN models, the variables were modelled by means of probability distributions; this allowed the authors to estimate uncertainty more accurately compared with other models other models [108–110].

Almuhaylan et al. [68] compared a MODFLOW model, three ANNs, and one adaptive neuro-fuzzy inference system (ANFIS) model developed in the Saq-Aquifer, Al-Qassim region (Saudi Arabia), an aquifer mainly characterised by medium-to-coarse sandstone. The modelling framework was implemented for assessing the impact of different groundwater pumping scenarios on aquifer depletion. The performance of ANN/ANFIS models for long-term future predictions of GWL and for finding a simple solution to the problem of undefined boundary conditions was examined. Deep learning models, e.g., recurrent neural network or convolutional neural network, are usually required for long-term predictions. The authors instead adopted a simple approach by changing the targets and predictions into GWL changes instead of GWL to develop a standard ANN/ANFIS simulation problem. Additionally, the training of the ANN/ANFIS model was handled with the prediction of changes in GWL instead of the direct simulation of GWL. The authors optimised the use of ANN model by choosing different combinations of architecture (number of hidden neurons and number of layers). The authors obtained a lower mean-square-error and a higher NSE in the training stage of ANN and ANFIS models compared with the calibration of the MODFLOW. Despite the hydraulic model being comparatively more reliable, ANN and ANFIS showed excellent performance, better than the MODFLOW model in terms of NSE. The authors did not simply remark any performance improvement of ANFIS with respect to ANN; they showed better performance in both with respect to the numerical model.

Chen et al. [63] applied a physically based model developed with MODFLOW and three ANN machine learning methods (ANN, RBF, SVM) to simulate the groundwater dynamics of the middle reaches of Heihe River, northwest China. The objectives were to assess the efficacy of machine learning models on reproducing groundwater dynamics in arid basins and to compare results of machine learning and numerical models to verify their applicability. The performance was evaluated by Root mean square Error (RMSE) and Coefficient of determination ($R^2$). As for the multilayer perceptron, the hyperbolic tangent sigmoid transfer function was applied in the neurons of the hidden layer and the linear transfer function was applied in the output layer; the number of hidden neurons was identified by trial-and-error procedure. Trial-and-error was used also to identify the number of hidden neurons for the RBF network. In RBF, the Gaussian radial basis function was applied in the neurons of the hidden layer and linear transfer function was applied in the output layer, respectively. As for the SVM, Gaussian function (i.e., radial basis function) was used as a kernel function to compute the Gram matrix. Furthermore, for each of the machine learning models, the ratio between RMSE in the prediction stage times RMSE the in training stage was calculated as a measure of the models' generalisation ability (GA). Machine learning models simulated historical data with higher performance with respect to numerical model, with the RBF model performing the best. In particular, SVM performed the best in the training stage, while RBF in the verification stage. Machine

learning models showed much less computation cost in training and prediction stages than those of numerical model in calibration and verification stages. However, because of the physical based mechanism, the numerical model showed a better generalisation ability. Therefore, authors concluded that machine learning models are applicable to problems that require a high number of model runs without considering the physical mechanisms (e.g., optimisations, real-time models, sensitivity/uncertainty analysis).

### 3.3. Testing Hybrid or Ensemble Models

Hybrid modelling approaches including data-preprocessing and/or combination of different machine learning techniques in different stages of the modelling have also been developed in the recent years to improve the efficiency of the machine learning methods [49].

Malekzadeh et al. [100] modelled the GWL in a well located in the arid agricultural area of Kabodarahang Plain (Hamadan, Iran) using MODFLOW and a hybrid artificial intelligence model. They compared an extreme learning machine model (ELM) and a combination of ELM with the wavelet transform (WA-ELM), intending to improve MODFLOW model calibration and optimise the prediction of GWL. Wavelet analysis is commonly executed for de-noising, compressing, and decomposing input data time series in the stage of data pre-processing. Similar to the Fourier transform, the Wavelet transform considers time series as a linear combination of multiple base functions, and has the ability to obtain time, frequency, and situation data simultaneously [111]. Malekzadeh et al. [100] divided time series into several sub-series using the discrete wavelet transform (DWT), and then used the decomposed components as input for the ELM model, instead of the main time series. Different families of the wavelet model were evaluated by comparing the values of R, RMSE, and BIAS, finding the mother wavelet used for the further steps. For each of the ELM and WA-ELM models, 10 different models were defined; the best-performing activation function and topology were chosen. As a result, the best models among the ELM and the WT-ELM models were selected. Then, the results of the hybrid method were compared to ELM and MODFLOW based on the MAE and RMSRE. They found that the WA-ELM model simulates GWL with higher accuracy with respect to both ELM and MODFLOW models.

Nikolos et al. [101] utilised ANNs to approximate a finite element model and combined it with a Differential Evolution algorithm (DE) to determine the best operational strategy for the productive pumping wells located in the northern part of Rhodes Island in Greece. A 3D finite-element simulation model of the study area was initially implemented using the Princeton transport code (PTC). The DE optimisation algorithm was successfully used for solving the optimisation problem, since it provides a solution close to the global optimum in a fully automated way. In the work of Nikolos et al. [101], the calls of the PTC model were replaced with an ANN in order to overcome the time-consuming integration of the PTC model within an evolution-based optimisation procedure. The training/evaluation data for an ANN model were produced by the PTC model. Several numbers of hidden nodes and training epochs were tested to adopt an optimum ANN topology. Then, the ANN was combined with the DE algorithm to solve two different water table elevation scenarios at the observation wells. The classic DE algorithm evolves a fixed size population npop that is randomly initialised [112]. After initialising the population, an iterative process was started which produces a new population until a given condition is satisfied. At each iterative step, a newly generated element can replace each element of the population. At the end of each run, the optimum solution was used as an input to both the PTC and the ANN models to test the accuracy of the ANN predictions and the effectiveness of the constraints.

The results of this procedure demonstrated that the ANN can be used as a quick surrogate model, providing very close to optimal solutions and allowing us to run an optimisation procedure with the DE algorithm in less than a minute instead of the several hours required to run the same process with the PTC model.

Sahoo et al. [102] proposed an ensemble modelling framework (Automated hybrid artificial Neural network, HANN) comprising spectral analysis (SSA), machine learning, and uncertainty analysis to facilitate improved GWL prediction with respect to computationally expensive physically-based MODFLOW models. The method was applied in two aquifer systems exploited for agricultural production (the Mississippi River Valley alluvial aquifer and the High Plains aquifer, USA), with the aim to clarify the influence of each climate variable on the irrigation demand and streamflow and predict groundwater level change. The best input for the ANN was selected by a hybrid data pre-processing method which includes: (i) decomposing the time series using SSA to extract significant reconstructed components (RCs); (ii) selecting the best RC of inputs by mutual information and genetic algorithm; (iii) and determining time lag components using cross-correlation analysis. Then, the simulations from the HANN model during the model testing period were summed to estimate the cumulative GWL change. The HANN results were compared to regional GWL simulations coming from MODFLOW models previously developed by many authors. HANN showed better performance in terms of MSE. The authors highlighted that the HANN shows a high model structure strength since it integrated a robust data pre-processing and input variable selection technique within the ANN model for capturing the impacts of the potential predictor variables on GWL change at observation wells.

Because the model is implemented and optimised for each well, they benefit from training values at each well. On the other hand, while showing a lower prediction error than the physical models, HANN cannot furnish the outputs typical of a physically-based model, such as water balance, residence time calculations, and flux estimates. Moreover, while a numerical model can be modified to include additional input or processes (e.g., supplied water), introducing new parameters would require the building of a new ANN model. Therefore, the authors concluded that each model type excels for certain applications.

Michael et al. [82] compared three machine learning techniques (DT, IDW, and ANN), which were used in a hierarchical approach, to improve GWL forecasting by combining data from different sources, including the results of a MODFLOW numerical model. They used a collection of prewritten modules (set up for each machine learning model) composed in a "data flow" program. The MODFLOW model is incorporated into the itinerary by creating a module that returns the head prediction by MODFLOW. A hierarchy of models was then arranged, with one model used to reduce the dimensionality of the largest data set (called "specialty model") and a second model ("expert model") trained with a combination of the remaining data and the specialty model results to obtain the optimum predictions. After linking together the modules into a machine learning itinerary, a model was automatically built by the itinerary from appropriate data sets to make predictions. At first, the hierarchical approach used machine learning models as both specialty and expert models; the results demonstrated that, based on mean predicted head errors, DT provided the best prediction among the machine learning models, while neural networks provided the least accurate prediction. The best machine learning model performed better than the MODFLOW model in terms of hydraulic head predictions computed across all observations used for calibrating the MODFLOW model. Furthermore, a very short time is required to train DT, and their simplicity allows quick planning of on-site adaptive field sampling. Interestingly, IDW showed a performance nearly as good as DT and IBW when using all of the data across time. The authors concluded that the accuracy of physics-based models can be improved by using a machine learning hierarchical approach in areas with substantial data. Using this method allows identifying (i) advantages and disadvantages of different machine learning approaches and (ii) which data are most significant for long-term monitoring objectives. Secondly, the MODFLOW model was used as a specialty model to test the potential for machine learning methods to automatically update existing numerical models. In many cases, such as groundwater remediation fields, it is not cost-effective to recalibrate numerical models whenever new data become available. Instead of updating existing models by tuning the parameters based on new data, physically based models

are considered as one source of knowledge about the site and are integrated to historical data using data-driven models. Results showed that, compared with the MODFLOW model predicted errors, the mean predicted head error and the standard deviation of predicted head error were consistently reduced with the best-combined model. This means that the model learns a compromise between numerical and machine learning models. Such combined hierarchies could allow an automated update of physically-based models, expanding and adapting the prediction as new data (but also analysis and modelling techniques) become available.

Fienen et al. [95] evaluated three machine learning techniques (BN, ANNs, and GBRT) to train models simulating the source of groundwater to several wells. The aim of the work was to predict local surface water impact due to new pumping wells. The regional 215,000 km$^2$ groundwater model of Lake Michigan Basin [113] impedes the evaluation of local-scale impacts due to the long runtime and the too-coarse grid. The solution was to emulate the groundwater flow model using a dataset of collocated numerical model input and output to build a statistical learning model ("metamodel", [114]), providing fast decision support to water managers which need to evaluate the permission to water abstraction. In practice, the numerical model was used to generate outputs reproducing several condition of the groundwater system; then, those outputs were used to train a statistical model, which could be subsequently used to make predictions without the need to run the regional model. The ability of the three techniques to extend MODFLOW predictions to areas with few samples was evaluated. K-fold cross validation (CV) was used to assess the models performance, as well as by hold-out data. The performance of the BN model (evaluated by means of R$^2$ and RMSE) was lower than the other two, and this could be due to the fact that the continuous input and output variables were both discretised into a small number of bins. All the three techniques can be implemented with commonly used commercial (in the case of BN) or open source (in the case of ANN or GBRT) software. The computational time is nearly instantaneous for all the three techniques while it takes longer to perform cross-validation. ANN or GBRT may be the best options for managers who need to achieve better predictive performance when a single response is considered. BN includes estimate of the uncertainty of predictions because all variables are treated as probability distributions. The authors concluded that the metamodelling approach is valid over a wide range of conditions and, as a screening approach, is helpful. A limitation of their approach is that it assumes that the response of the system to pumping rates is linear; thus, this assumption is violated at high pumping rates.

Miro et al. [99] presented a hybrid empirical–dynamical approach application of machine learning models to a Robust Decision Making study to evaluate the effect of groundwater managed recharge. They developed an empirical model representing a high-resolution MODFLOW model previously set-up in two basins located in a drought-prone region of the American West: the San Bernardino and the Rialto-Colton basins, San Bernardino Valley Municipal Water District (Valley District, U.S.). Inputs (recharge, pumping) and outputs (resulting head) of the MODFLOW model were used to train three machine learning methods (Random Forest, Support Vector Machine regression, and Artificial Neural Network) to predict the annual change in GWL. Then, the ability of machine learning methods to simulate the output of the MODFLOW model was assessed to investigate which model is capable of reproducing the best average basin conditions. Based on R$^2$, the most accurate results were obtained with RF. The authors concluded that RF is able to reproduce time series trends in GWL as well as capture the variability in MODFLOW model predictions. In that way, the authors obtained a significant reduction of computational time: each MODFLOW run without the RF model would have taken approximately 36 years in a standard computing environment, instead of 24 h while simulating MODFLOW with a RF representation of the groundwater system. The procedure is integrated in a Robust Decision Making (RDM) process: the novel application of machine learning represents an improvement to the field of decision-making under deep uncertainty that allows reducing

computational times and permits a greater exploration of the uncertainty space, such as future climate changes and drought conditions.

### 3.4. Reducing and Correcting Model Errors by Means of Machine Learning Approaches

Despite a correctly constructed and calibrated groundwater model being able to furnish valuable information about the system behaviour, the unaccounted uncertainty, which is typically associated with the phases of model development and parametrisation, can result in large localised simulation errors.

Xu et al. [79] tested two machine learning techniques (Instance-based weighting and support vector machine regression) to correct the prediction of two physically-based models, successfully improving the head prediction accuracy. The authors applied the error-correcting data-driven models to temporal, spatial, and spatiotemporal prediction. The core of the study relies on the selection of historical residuals of the physically-based models, which were used to train the data-driven models. Then, the physically-based model was used to make predictions, and the trained data-driven models were used to predict the error of the predictions. Finally, the updated head was obtained by adding the predicted error to the head simulated by the physically-based model. The procedure was applied to two real-world groundwater flow models having different data densities and extents of temporal and spatial structures in the error. The first is the regional Republic River Basin (RRCA), covering portions of eastern Colorado (USA), a 79,396 km$^2$ model [115] developed to resolve water conflicts as growing water demand led to dramatically increased groundwater pumping. The second is the Spokane Valley–Rathdrum Prairie aquifer (SVRP) (USA), an 844 km$^2$ aquifer subjected to groundwater pumping stresses. The two models differ in various aspects, including parametrisation, calibration, grid resolution, data density, and calibration strategy, leading to different spatial patterns in model residuals.

In the case of RRCA, data were pre-processed by cluster analysis: for temporal prediction, observation wells were clustered using the agglomerative hierarchical clustering algorithm according to their spatial location. In spatial and spatiotemporal prediction scenarios, input data were clustered by the k-means algorithm. Each cluster was subdivided into a training and a validation dataset, and data-driven models were applied to each subset.

In the case of the SVRP model, cluster analysis was not implemented because residuals did not show local patterns; thus, the data-driven models were applied only to the temporal prediction scenario. In the same way, to the RRCA case study, IBW and SVR models were built to forecast the error of the simulated head taking as input features the well location and MODFLOW computed head; then, the updated head was computed. For both case studies, five-fold CV was used to adjust the parameters of IBW and SVR.

The magnitude and biasedness of the prediction error (evaluated by means of ME and RMSE) were sufficiently reduced. The authors found that this complementary modelling framework was computationally efficient. New data can be easily incorporated into the training dataset. Therefore, data-driven models can be used to improve the prediction of the physically-based model for long-term prediction and under conditions different from the one used during calibration. A limitation of this methodology is that it applies only to physically based groundwater models with epistemic errors in the simulation results, while it is not suitable for models with calibration error following Gaussian distribution with zero mean and variance comparable with the observation error.

Demissie et al. [85] developed a complementary approach that integrates the calibrated groundwater MODFLOW model with data-driven models to detect and predict systematic errors in groundwater model simulation in a hypothetical test case based on the Argonne National Laboratory, Illinois (USA), a site affected by groundwater contamination by radioactive substances and volatile organic and with phytoremediation installed to clean up the soil. Using the groundwater model residual analysis results, the authors implemented four data-driven models (ANN, DT, SVM, and IBW) for simulating and correcting the groundwater head predictions both in time and in space. The data-driven models were then

used to update the head predictions. The updated models showed improved performance compared to the MODFLOW head predictions at all observation wells in terms of RMSE reduction. ANN performed better in updating the future predictions but required a longer time to train the model and the definition of many parameters. IBW updates showed a better performance in the case of spatial prediction, probably because the number of spatial data was too small for the other three models to learn the spatial patterns of the residuals.

## 4. General Results

The general outcomes derived from the 16 reviewed studies are discussed, such as the results related to the key area of model use, input variables, simulation period of physically-based models, time step, dataset size and division, and software used.

### 4.1. Key Area of Groundwater Model Use

In general, machine learning models are developed to achieve a better performance in GWL forecasting in areas where groundwater management strategies are strictly required to ensure proper resource availability while protecting the environment and groundwater related ecosystems. This is especially needed in areas where the aquifers have been overexploited; where the groundwater recharge is scarce (drought-prone regions); and in coastal areas, where groundwater is threatened by saltwater wedge intrusion. Most of the reviewed papers (four, Figure 3) concern water planning and supply, usually at the catchment scale. A minor number of papers (three) focus on the groundwater management in farming systems; in coastal waters; in drought-prone regions. In two cases, machine learning models are developed in areas with contaminant pollution and phytoremediation plants. Finally, one paper attempted to use machine learning models to represent the impact of groundwater abstractions on river discharge across a wide range of conditions. From the reviewed papers, it is not possible to recommend a machine learning technique for a specific key area of model use. ANN is the most-used technique in the case of water planning and supply (also in drought-prone areas), followed by BN and SVM (Table 1).

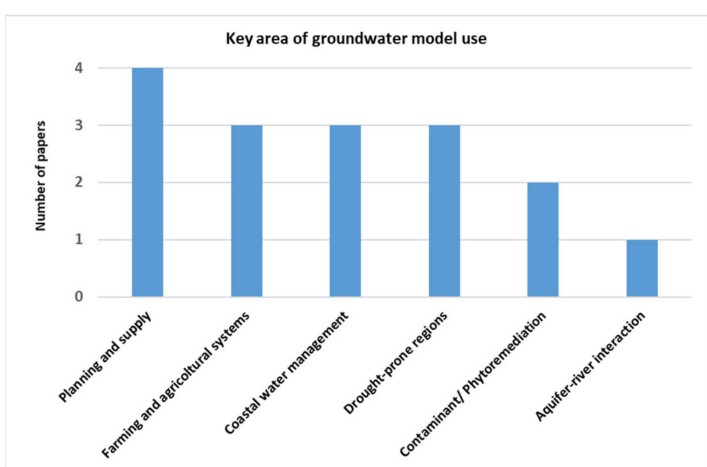

**Figure 3.** Key areas of groundwater models use in the reviewed papers.

### 4.2. Input Variables Employed for Machine Learning Modelling

Figure 4 shows the input variables that have been utilised in machine learning modelling. The past GWL time series are the most frequently used input variables to predict GWL; among 16 papers, 13 employed the GWL as an input variable. The precipitation or the net precipitation (i.e., the recharge) has been frequently used (four times the rainfall and five times the recharge, for a total of nine times) as an input variable. Moreover, other hydrological time series (e.g., pumping rates, temperature, evapotranspiration) have been also employed as the input variables in the reviewed papers. Since machine learning models can work with any data, there are many other input variables which have been

used once in the reviewed papers, even if with less frequency (i.e., aquifer discharge, dew point, river stage, river flow depletion, irrigation demand). It is worth noting that the input data are commonly selected based on the availability of data rather than a physical analysis of the system. In particular, the degree of accuracy of the prediction will depend upon the spatial and temporal resolution of the monitoring network from which the model is developed for making predictions. Thus, the choice of input variables is often driven by the availability of proper time series.

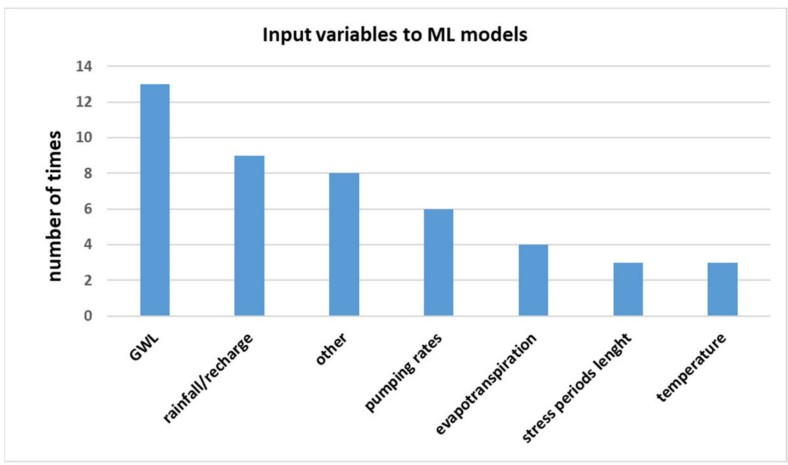

**Figure 4.** Input variables employed for the machine learning models.

### 4.3. Simulation Period of Physically-Based Models

The simulation period of the physically-based models varies from 141 to 1 year; nine physically-based models adopted a simulation period between 1 and 15 years; five physically-based models used a run simulation period between 16 and 35 years; two physically-based models adopted a run period higher than 35 years. In the reviewed papers, no mention was paid to a direct relation between the simulation period length of physically-based models and prediction accuracy of machine learning models. The size and layering of original numerical models ranges from 204,764 km$^2$ and 20 layers [95] to 0.75 km$^2$ and 1 layer [55]. Results suggest that there are no machine learning techniques nor groundwater management problems specifically suitable for a given range of physically-based size.

### 4.4. Time Step

The majority of the reviewed papers (8 among 16) used monthly time step for the machine learning simulations, followed by daily and weekly (both used in two papers) and quarterly (one paper). The time step selection was not declared in three of the reviewed papers. The frequent choice of monthly time steps is probably justified by the large availability of monthly recorded GWL data compared with other time steps. However, daily time steps are needed when modelling local-scale problems, such as river–aquifer interaction, or in some coastal water problems, where GWL are influenced by the tidal effects which induce daily variation to GWL.

### 4.5. Data Set Size

The number of total data used for groundwater modelling is highly variable. In three papers, only the number of wells was specified, without reporting the number of measure for each well. Among the papers which declared the size of data, the data set ranges from 300,000 sets [79] to 23 sets [96]. There is not a range of data set size which was more commonly used: five models used a number of dataset from 23 to 301; five models used from 1872 and 4911 datasets; four models used from 11,088 to 300,000 datasets. There is not a direct proportion between area of the physically-based model and data sets. Usually, smaller data sets are associated with a smaller size of the physically-based model. However,

in some cases, large extent models (which is, larger than 10,000 km$^2$) are covered by a relatively small number of data (e.g., ref. [3]). There is not any recommendation in the reviewed papers about the density of samples which optimises the model performance. However, denser distributed training data allow achieving the best performance in temporal prediction scenarios. For example, the ANNs' ability to learn or generalise system behaviour is limited by the data with which it is trained. Machine learning models can fail to accurately predict GWL in areas where a scarce number of data for training is available, and results can be worse than those of numerical models.

### 4.6. Subset for Machine Learning Model Training, Validation and Testing

As explained in Section 2.2.1, the data available for modelling are subdivided into a training dataset (used during the learning phase of the machine learning model to produce a function representing the system behaviour) and into a testing dataset (used to evaluate the model's performance). Some authors subdivide data in three groups: training, testing, and validation; validation aims to check the model's prediction ability with a new input dataset.

There is not a specific rule for determining the optimum percent of data division for training, validation, and testing tasks. However, it can be noted that in all cases (except Sahoo et al., 2013 [102]) the dataset for training in the reviewed papers was always at least 60% (Figure 5), reaching 95%. In the majority of the papers (9 among 16), the percent of training dataset exceeded 80%. With regard to the testing dataset, authors use a percentage highly variable, between 4.5% and 40%. Only three of the reviewed papers used three subsets for training, for testing, and for validation, respectively. In these cases, the main subset was used for training (60%, 69%, and 52%), and the remaining data were equally distributed between testing and validation subsets or subdivided into 30% and 18% for testing and validation, respectively. In Banerjee et al. [96], the division into validation or testing sets was not mentioned, and the performance criteria were only mentioned for the training data, as already reported in ref. [19]. It can be concluded that a robust machine learning model should always be based on at least 60% of the training data, and 40% of the testing data.

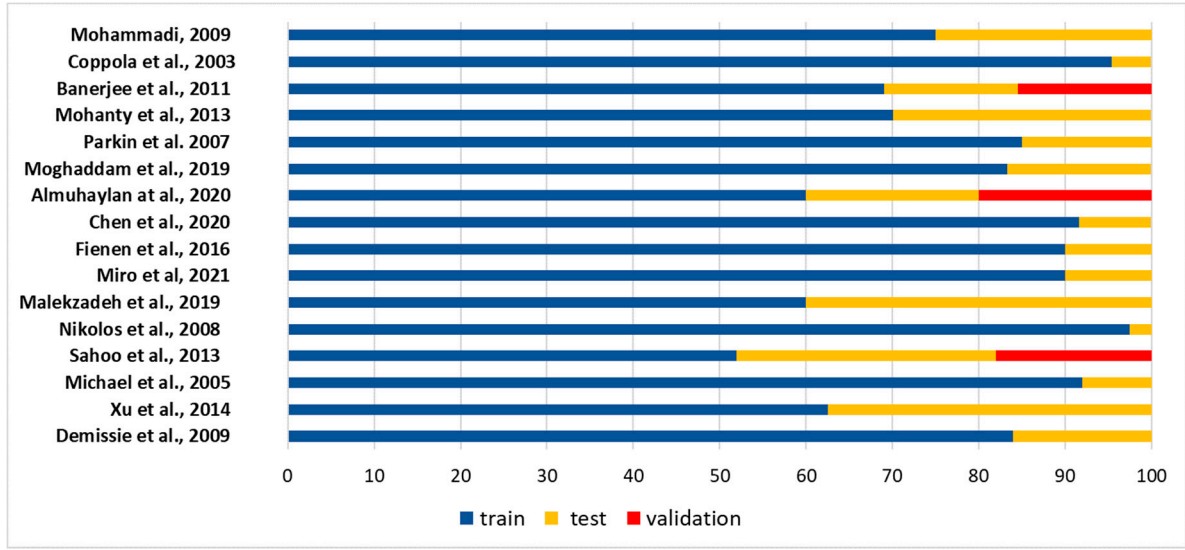

**Figure 5.** Percentage of the training and testing datasets used in machine learning modelling. Data from: Mohammadi, 2009 [59], Coppola et al., [3], Banerjee et al., 2011 [96], Mohanty et al., 2013 [97], Parkin et al., 2007 [98], Moghaddam et al., 2019 [76], Almuhaylan at al., 2020 [68], Chen et al., 2020 [63], Fienen et al., 2016 [95], Miro et al., 2021 [99], Malekzadeh et al., 2019 [100], Nikolos et al., 2008 [101], Sahoo et al., 2013 [102], Michael et al., 2005 [82], Xu et al., 2014 [79], Demissie et al., 2009 [85].

*4.7. Used Software*

Table 2 shows the number of times that each software was used to develop the ML models. It should be noted that in 11 cases, the software was not mentioned. Matlab is the most used software (nine times): when mentioned, the specific toolbox which adapts to the models' purpose is reported. In the hybrid approach of Michael et al. [82], the data-to-knowledge D2K software was used, a java-based data mining tool from the National Canter of Supercomputing Applications which allows for graphic data flows [116]. The results of this review indicated that Matlab can be easily used to implement the machine learning models; the variability and flexibility of its toolboxes clearly represent an advantage. However, the modellers can choose a range-free software with comparable skills.

**Table 2.** Software used for the machine learning models in the reviewed papers.

| Machine Learning Model | Software | Commercial/Free | n of Times |
|---|---|---|---|
| ANN | Matlab | c | 3 |
| | R-neuralnet package | f | 1 |
| | LINGO | c | 1 |
| | not specified | | 7 |
| RBF | Matlab | c | 1 |
| ANFIS | Matlab | c | 1 |
| TLRN | NeuroSolution | c | 1 |
| ELM, WA-ELM | Matlab, Matlab wavelet toolbox | c | 1 |
| BN | Hugin Lite 8.3; | c | 1 |
| | netica Software, CVNetica (for cv) | c | 1 |
| IBW | Matlab Statistic Toolbox TM | c | 1 |
| | not specified | | 1 |
| SVM | Matlab Statistic Toolbox TM | c | 2 |
| | not specified | | 2 |
| DT | not specified | | 1 |
| RF | R (randomForest package) | f | 1 |
| | not specified | | 1 |
| GBRT | Phyton (scikit-learn library) | f | 1 |

## 5. Specific Results

This section aims to furnish specific information to orient modellers choosing the appropriate machine learning approach based either on the properties of each of the examined model (e.g., the most used algorithms, model structure, tuning parameters: Section 5.1) or on advantages and disadvantages arising from the comparison between different machine learning techniques (Sections 5.2–5.4).

*5.1. Properties of the Machine Learning Techniques Used in the Reviewed Papers*

This section describes the results of the assessment of the machine learning techniques mostly used in the reviewed papers: ANNs, RBF, ELM, BN, SVM, DT.

Artificial Neural Networks

An assessment of the reviewed studies on ANNs revealed the following issues:

- Feed-forward multilayer perceptron with a backpropagation learning algorithm was the most used ANN technique in the reviewed papers.
- The training algorithms used in the reviewed papers were Levenberg Marquardt, Bayesian regularisation, scaled conjugate gradient, quick propagation algorithm, back-propagation algorithm, and resilient backpropagation. The most used were Levenberg Marquardt [60,117], which integrates the advantages of two training algorithms,

namely the steepest descent, and Gaussian–Newton methods, and searches for the global minima function to optimise the solution [68]; some authors point out that this is the less time-consuming algorithm.

- The transfer functions used for the hidden layer are: sigmoid, sine, hardlim, triangle basis, radial basis, hyperbolic tangent, linear, and logistic.
- The most common structure of ANN in the reviewed papers is a feed-forward ANN with a single hidden layer, with sigmoid transfer function in the hidden layer and linear transfer function in output layer. The best structure and number of hidden neurons are chosen by trial-and-error or cross-validation.
- The final structure of multilayer perceptron is usually chosen as the one resulting in minimum error and maximum efficiency during training.
- ANNs are capable of achieving substantially higher predictive accuracy at observation wells than the physically-based numerical model, with fewer inputs and lower developmental effort and cost. The choice of the appropriate training data size is a key issue; it should be evaluated considering many aspects, such as the required model accuracy, the number of connection weights, the complexity, and the level of noise in the system [3]. Moreover, it is important to find the optimal ANN topology ensuring satisfactory generalisation capability for any given problem. This is generally achieved by testing different topologies and transfer functions.

Radial Basis Function

Chen et al. [63] applied the Gaussian radial basis function to the neurons of the hidden layer and the linear transfer function in the output layer. RBF showed a better predictive performance and its computation cost in training and prediction stages were much less than those of numerical model in calibration and verification stages.

Extreme Learning Machine

In the work of Malekzadeh et al. [100], the number of hidden neurons for the ELM model was optimised by trial-and-error; results showed that model prediction was not significantly improved by increasing the number of hidden layer neurons. The sigmoid activation function provided higher simulation accuracy. The advantages of ELM with respect to other models are its modelling simplicity, easy coding, and quick computation for simulations in complex systems.

Bayesian Network

In the work of Moghaddam et al. [76], a BN structure was built, generating 108 possible states. The input parameters included rainfall, GWL in the previous month, average temperature, aquifer recharge, and discharge. The performance of BN models was evaluated by means of the $R^2$ and RMSE derived for all the observation points. In Fienen et al. [95], the BN was implemented with variables that were supposed to have the greatest influence on the source of water to wells: the distance to surface water, the surface water percent, the distance of 1st-order stream, and the percent of 1st-order stream. The continuous values of variables were discretised into bins; this permits performing predictions as discrete conditional probabilities without requiring a priori assumptions about distributions. Both the number of nodes and the number and ranges of bins were adjusted by 10-fold cross validation, and the set of parameters resulting in highest $R^2$ was selected as the optimal model.

Support Vector Machine

The most used kernel function with Support Vector Machine technique was the Gaussian Radial Basis Function, although several functions were tested (linear, radial bias, sigmoid). Cross-validation was the most used method for the optimisation of parameters (i.e., gamma value for the radial basis function and the regularisation coefficient), although Chen et al., 2020 [63] used Sequential minimal optimisation.

Decision Tree

Three types of decision trees were used in the reviewed papers: decision trees, random forest, and Gradient-Boosted Regression tree. In Demissie et al. [85], k-fold cross-validation was used to optimise the DT's pruning levels (used to reduce the complexity of the trees and

reduce overfitting by removing redundant trees sections). Michael et al. [82] highlighted that decision trees have the ability to incorporate different data sources; thus, existing historical data can be combined with new surrogate or indicator data (such as rainfall) to detect whether the new data indicate potential problems that would warrant the collection of more traditional samples. To note, while DT provides the most accurate prediction improvement with updated data, IDW represents a good compromise between prediction accuracy and easy implementation. In the Random Forest model of Miro et al. [99], the parameters to optimise were the pruning levels, the learning rate and maximum tree depth, and the number of trees examined. Hyperparameters were adjusted generally by cross validation. Furthermore, the RF with the number of trees providing sufficient performance with a reasonable computational time was chosen as best model. The main advantage of using RF model is the reduced computational time with respect to numerical models, which allows incorporating it as a step of decision-making studies to speeds up the process. In the Gradient-Boosted Regression Trees [95] the parameters defining the individual trees included tree depth, shrinkage (a form of regularisation), learning rate, and maximum number of leaves on a tree. One advantage of GBRT is the possibility to use a variety of loss functions; Fienen et al. [95] used the HUBER loss function, an intermediate between squared difference and absolute difference. Hyper parameters were adjusted by cross validation with k = 10. The key tuning hyperparameters were the learning rate and maximum tree depth. The tradeoff curves of the best set of tuning parameters were explored for each technique; other metrics of skill/fit were calculated based on $R^2$ score.

*5.2. Comparison between Machine Learning Techniques*

The comparison between different machine learning techniques in the reviewed studies showed that:

-   The performance of ANN with RBF as the activation function performed the best in simulating groundwater dynamics in arid basins, compared with ANN multilayer perceptron and SVM [63]. In detail, SVM performed the best in the training stage, while RBF in the verification stage; ANN's performance was lower than these two.
-   Regarding ANFIS, no improvements are remarked with respect to ANNs, although greater performance with respect to the MODFLOW numerical model is documented [68].
-   With respect to multilayer perceptron ANN, TLRNs can provide an appropriate tool for processing time-varying information. The main advantage is that TLRNs require a lower memory compared to multilayer perceptron, due to their lower network size. Furthermore, TLRNs have a low sensitivity to noise.
-   Compared to simple ANN, ELM showed better performance, much less modelling time, less modelling error, and less weights norm [100].
-   With respect to ANN, BN models provided easier implementation, higher prediction accuracy, and a greater ability to deal with missing or incomplete data [46]. It allows an uncertainty estimation more accurate than other machine learning models because the variables are modelled by means of probability distributions. When used as a metamodel, replacing a regional groundwater model to simulate the source of water-to-well [95], BN showed lower cross validation predictive skill compared with ANN and GBRT. However, the BN includes estimates of the uncertainty of predictions as part of the technique. GBRT required the least time with respect to BN and ANN. Thus, in this case, the choice between a statistical learning approach such as ANN or GBRT and the BN approach depends upon the preference of the modeller and the aims of the problem.
-   When used to predict the annual change in GWL as effect of managed recharge, RF produced the most accurate average basin GWL representation respect to observations, compared with SVM and ANN [99].

*5.3. Results of Testing Hybrid or Ensemble Models*

The use of hybrid models and a combination of techniques for data pre-processing (described in Section 3.3) allowed a significant improvement in each modelling phase.

- ELM and WA-ELM were both used to simulate GWL in an arid basin [100]. However, the ELM model with the db2 mother wavelet for data pre-processing showed a better performance with a significant accuracy improvement compared with the physically-based models.
- The hybrid approach of Nikolos et al. [101] provides a fast way to integrate the physically-based models within an evolution-based optimisation procedure (DE algorithm) by replacing the calls of the PTC model with an ANN. The ANN provides a tool to perform an optimisation run with the DE algorithm with very short time, serving as a fast and accurate surrogate model.
- The hybrid modelling approach HANN [102] showed a high model structure strength since it integrated a robust data pre-processing and input variable selection techniques.
- Using machine learning models in hierarchical approach can significantly improve the results of physics-based models [82]; moreover, by that way, advantages and disadvantages of different machine learning models are identified and insights are provided into which data are most valuable to long-term monitoring objectives and which are not. In particular, Michael et al. [82] found that DT consistently provided the most accurate predictions of hydraulic head compared with IDW and ANN. However, when using all of the data across time, IDW showed substantial improvements. Given that IDW is simple to use and is widely accepted among practitioners, it could be considered as an optimum choice.
- The computational time of regional physically-based models can be substantially reduced by introducing an empirical (or statistical) representation of numerical models; this consists of machine learning models trained using numerical models inputs and outputs, which can be used to make predictions of variable of interest [95,99].

*5.4. Results of Machine Learning Models Used to Reduce or Correct Errors in Physically-Based Models*

This section summarises the main features of the machine learning models used for error correction and reduction (described in Section 3.4). IBW models were constructed to correct MODFLOW models by using the position of observation wells, calculated heads, evapotranspiration rates, and stress periods as inputs, and the residuals of MODFLOW model as outputs [79,82]. The parameters to optimise were the values of weighting function parameters and the number of neighbors n. Parameters of SVM models were already described in Section 5.1. When used to correct the error of physically-based models, both IBW and SVM have been shown to successfully reduce the magnitude and biasedness of the prediction error. Xu et al. [79] remarked that the popularity of SVM can be attributed to: (1) good generalisation performance; (2) always having a globally optimal solution (instead of local optima); (3) representation of the solution sparsely by a small subset of all training examples (Support Vectors) [87]. On the other hand, because IBW does not involve the training process and is less affected by the size of the training dataset, it is particularly recommended when the number of data is too small for other techniques to learn the spatial pattern of residuals [85]. In the case of spatial prediction, the simple IBW updates the future predictions better than DT and ANN and SVM; IBW models allow locally improving the results, and its degree of localisation and complexity can be adjusted flexibly. Thus, when groundwater model errors show local patterns, the application of IBW is advantageous. When considering both spatial and temporal prediction, IBW performed roughly as well as the more sophisticated SVM.

## 6. Discussion

Assessments of machine learning applications in GWL forecasting reveal that the performance of such methods is comparable to, or even more accurate than, that of numerical ones. Overall, the reviewed papers prove the capability of machine learning methods for

capturing the nonlinear relation between groundwater and climate variables, especially where physically-based models would be difficult to implement. Machine learning models require a lesser number of input parameters and avoid the model building and parameter estimation stages typical of numerical models. Machine learning models can be a valid alternative for numerical models requiring long runtimes (i.e., complex regional models, models simulating many different processes, uncertainty analysis, sensitivity analysis), being capable of reducing computational times without sacrificing accuracy of detail in GWL forecasting. The very short time allows integrating machine learning models in decision-making processes when rapid and effective solutions for groundwater management need to be considered. Data-driven models are computationally efficient tools to correct head error prediction of numerical models; they work for error from multiple sources, and do not invoke assumptions on the error distribution [49]. Input data different from those used in the training stage can be included (e.g., pumping rate, boundary conditions, etc.); therefore, the data-driven models can be used to improve the prediction of physically-based models under scenarios that differ from the conditions used for calibration. Moreover, machine learning models can be applied successfully for modelling river–aquifer interactions.

Many studies exist concerning the use of machine learning models for groundwater simulation, developed on the basis of a limited number of observation points, without comparing results with numerical models. Conversely, the comparison of numerical and machine learning models is still a scarcely diffused task. In these comparative studies, each modeler uses the machine learning techniques for fixing a specific weakness of the numerical model, or to ameliorate poor fitting between simulated and observed values; in most cases, modellers explore different machine learning techniques to establish which one adapts better to its scopes. However, there are currently no well-defined procedures for the use of machine learning techniques to enhance results of numerical models, and this can limit the diffusion of the method. Another reason can be that the modeler should be familiar with both numerical and data driven models to correctly use both model types. Indeed, even if machine learning modelling does not consider the behaviour of the natural system, a certain degree of knowledge about the hydrological parameters and how they affect the results is required in order to avoid, for example, model overfitting (which means fitting the model to all the input parameters, preventing the generalisation ability of model, which is, in turn, given from the parameters effectively influencing groundwater level). In other words, the modeler should be able to manage both physically based data and statistical distributions of data, coupling different skills: those typical of hydrogeologists and those typical of statisticians/mathematicians. In many cases, a modeler (or a team of modellers) can meet both these requirements, but it is not so common. In addition, machine learning models are viewed with some skepticism by numerical modellers. Physically based represent the technique most widely diffused and used by local administrators for groundwater management. Usually, the results of a physically based model are improved by the integrating new observations (when available) or by tuning model parameters in order to modify the conceptual model. The machine learning approach, instead, aims at detecting the inherent mechanism, increasing prediction skills without deriving this from physical knowledge. This 'black box' nature, where no insight is gained into how the model generated the solution, is not widely accepted among numerical modellers and can prevent the use of machine learning models.

Regarding different machine learning methods to simulate the GWL when numerical models already exist, it can be said that from this review it is not possible to make a recommendation about one particular type of machine learning model for a specific problem. One advisable option could be testing different types of machine learning techniques in the different phases of the GWL modelling to detect the proper machine learning method in each stage and then couple them to achieve an optimum performance. However, hybrid modelling such as the combination of different techniques (e.g., data pre-processing such as time series decomposition or spatial clustering) and the hierarchical combination of machine learning models help to improve the accuracy of prediction. Moreover, some

of the machine learning models appear to be suitable for updating numerical models previously calibrated, improving predictions as new data are collected (i.e., DT, [82]; IBW and SVM, [79]). Furthermore, when using machine learning models to correct the error of physically-based models, both IBW and SVM show better performance than DT and ANN. However, the simple IBW allows locally improving the results, and this suggests that it is suitable when errors show local patterns.

Some authors highlighted the main disadvantages of machine learning models with respect to numerical models:

- The numerical models are comparatively more reliable. While showing a lower prediction error than the physical models, machine learning models cannot return many of the outputs of a physical model, such as flux estimates or total water balance.
- Xu et al. [79] found that data-driven models are difficult to interpret physically. The updated head no longer conserved mass for the given model inputs, which can confound the physical interpretation of the results and prevent understanding errors in the conceptualisation of the groundwater system.
- Numerical models exhibit a higher generalisation ability than machine learning methods because they are based on the physics of the system [63]. Conversely, machine learning models are applicable to problems that require a high number of model runs without considering the physical system (e.g., optimisations, real-time models, sensitivity/uncertainty analysis).
- Usually, while the machine learning models may be more efficacious for predicting short-term GWL and reproducing highly localised flow impacts, numerical modelling is more appropriate for long-term projections, or in areas where field data are insufficient for the given problem. However, it should be remarked that Almuhaylan et al. [68] were able to use machine learning models to perform long-term prediction (up to 50 years), by training the ANN/ANFIS model for the prediction of changes in groundwater levels instead of the direct simulation of water levels.

Thus, each type of model (numerical or machine learning) is suitable for a specific type of problem. As suggested by many authors, numerical and machine learning models can be successfully used as complementary to each other as a powerful groundwater management tool:

- when few field data exist, the results of numerical models can be improved by training machine learning models, which allow to obtain accurate groundwater level forecasting at specific observation wells;
- machine learning models cannot substitute a numerical model as one single model, but can be used to simulate water table fluctuation at every individual observation well with reduced computational time;
- accurate results of machine learning models in specific test sites can be used to obtain the best GWL data required by the numerical model as input;
- the physical dynamics of the system must be sufficiently understood by the modeller in order to identify the important predictor input variables of machine learning models. Results of numerical models help to understand the physical system; this can help, in turn, choosing the input parameters for machine learning models. Coppola et al. [3] suggested using ANNs to perform a sensitivity analysis on the interrelationships between input and output variables;
- Numerical models can simulate different scenarios, allowing for detection areas requiring particular management strategies, thereby supporting the design of an effective monitoring network, which, in turn, may improve both machine learning predictive capability and performance.

Given the results of this review, one should evaluate the best machine learning technique based on:

- The aim of the work, for example: improvement of prediction at some well location, numerical model error correction, numerical model updating;

- the need to produce a probability distribution of the results and obtain uncertainty estimation within the model, (i.e., in areas with few data);
- the availability of data for training and testing (number and spatial-temporal distribution);
- the need to speed up decision making processes and reduce the computational time;
- the degree of expertise of the modeller, which should drive the searching for a good compromise between model complexity and prediction performance.

To note, this review only accounts for groundwater flow models; robust groundwater flow models are the basis for setting up groundwater solute transport models. The comparison between physically based and machine learning models focused on groundwater solute transport should be the subject of future research.

## 7. Conclusions

This study presents a review of 16 papers regarding the use of numerical models and machine learning techniques for the prediction of groundwater level, which were published in 10 international journals and 1 book from 2003 to 2020. Machine learning techniques are used to improve or speed the prediction process of physically-based models, which are developed with different codes and software, from regional to site scale, and with data collected over time windows spanning from one to hundreds of years. Machine learning methodologies, approximating the complex behavior and dynamics of physical systems, allow for the optimisation of predictions of a large number of scenarios within a short period of time, compared with the long computational time required for the corresponding simulation time using a numerical model. Machine learning models do not return many of the outputs of a physical model, such as flux estimates and residence time calculations, or total water balance. Thus, machine learning models cannot be used to substitute numerical models in large study areas, but are affordable tools to improve predictions at specific observation wells. Results of this review suggest that numerical and machine learning models can be successfully used as complementary to each other as a powerful groundwater management tool. The machine learning techniques can be used to improve the calibration of numerical models, whereas the results of numerical models allow understanding the physical system and selecting proper input variables for machine learning models. Among the machine learning techniques, the hybrid machine learning models show better results accuracy.

**Funding:** This research received no external funding.

**Acknowledgments:** Author is very grateful to the reviewers for their interesting comments, which allowed to improve the quality of the manuscript. A special thank also goes to Daniel Feinstein, for his precious suggestions and support.

**Conflicts of Interest:** The author declares no conflict of interest.

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
