# Peer review of "Improving Results of Existing Groundwater Numerical Models Using Machine Learning Techniques: A Review"

_water, doi:10.3390/w14152307_

Round 1

Reviewer 1 Report

1)      This paper should be reorganized. First, The section “Introduction” has only one paragraph which should be improved. Second, Section 4 is divided into 11 sub-sections. So many sub-sections make this section difficult to understand. Section 4 should be improved to show the main objective.

2)      Section 2: the characteristics of each numerical model and machine learning model should be discussed, such as the merit and demerit.

3)      Figures and tables: The quality of the figures and tables should be improved, especially Table 1.

4)      There are so many abbreviations made this paper difficult to read.

5)      The author concluded the numerical models and ML techniques can be successfully used as complementary to each other as a powerful groundwater management tool. But the reason for the conclusion is not sufficient. Firstly, the author only gave 16 reviewed papers. If the integrated method of numerical model and ML techniques is better, why there are so few studies on the integrated method. The author should give more explanation. Furthermore, the comparison of the simulated result, such as groundwater level, between the integrated method and the simple numerical model is benefit to support the conclusion.

Author Response

 Dear reviewer

please find here my answer to your comment. I am grateful to you for giving me the opportunity to improve the manuscript.

#1 QUESTION: This paper should be reorganized. First, The section “Introduction” has only one paragraph which should be improved

#1 ANSWER: According with the reviewer’s suggestion, I improved the literature in the introduction, adding reference about application of machine learning models in different disciplines (see also comment #5 by reviewer 2), and other references throughout the section (e.g., about application of machine learning techniques in hydrology, water resources, machine learning models performance compared  to traditional statistical models, etc), in particular:

Hydrology:

  • Deka, P.C. Support vector machine applications in the field of hydrology: a review. Applied soft computing 201419, 372-386.
  • Lange, H., Sippel, S. Machine learning applications in hydrology. In Forest-water interactions;  Levia, F.,Carlyle-Moses, D. E., Lida, S., Michalzik, B., Nanko, K., Tischer,  A., Eds.,Ecological Studies, Springer Nature, Cham, Switzerland, 2020; Volume 240, pp. 233–257.  https://doi.org/10.1007/978-3-030-26086-6_10
  • Rasouli, K., Hsieh, W. W., Cannon, A. J. Daily streamflow forecasting by machine learning methods with weather and climate inputs. Journal of Hydrology 2012414, 284-293.
  • Xu, T., Liang, F. Machine learning for hydrologic sciences: An introductory overview. Wiley Interdisciplinary Reviews: Water 20218(5), e1533.
  • Yaseen, Z. M., Sulaiman, S. O., Deo, R. C., Chau, K. W. An enhanced extreme learning machine model for river flow forecasting: State-of-the-art, practical applications in water resource engineering area and future research direction. Journal of Hydrology 2019569, 387-408.

Water resource:

  • Solomatine, D. P. Applications of data-driven modelling and machine learning in control of water resources. In Computational intelligence in control; Mohammadian, M., Sarker, R. A., Yao X., Eds., Idea Group Publishing, Hershey, PA, USA, 2002; pp 197–217.
  • Yan, J., Jia, S., Lv, A., Zhu, W. Water resources assessment of China's transboundary river basins using a machine learning approach. Water Resources Research 201955(1), 632-655.

General machine learning models features:

  • Besaw, L. E., Rizzo, D. M., Bierman, P. R., Hackett, W. R. Advances in ungauged streamflow prediction using artificial neural networks. Journal of Hydrology 2010, 386(1-4), 27–37. https://doi.org/10.1016/j.jhydrol.2010.02.037
  • Daliakopoulos, I. N., Tsanis, I. K. Comparison of an artificial neural network and a conceptual rainfall–runoff model in the simulation of ephemeral streamflow. Hydrological Sciences Journal 2016, 61(15), 2763–2774. https://doi.org/10.1080/02626667.2016.1154151
  • Piotrowski, A. P., Napiorkowski, J. J. A comparison of methods to avoid overfitting in neural networks training in the case of catchment runoff modelling. Journal of Hydrology 2013476, 97-111.
  • Kingston, G. B., Maier, H. R., Lambert, M. F. Calibration and validation of neural networks to ensure physically plausible hydrological modeling. Journal of Hydrology 2005314(1-4), 158-176.

Comparison between machine learning model performance and traditional statistical approaches

  • Elith, J., Leathwick, J. R., Hastie, T. A working guide to boosted regression trees. Journal of Animal Ecology 2008, 77(4), 802–813. https://doi. org/10.1111/j.1365-2656.2008.01390.x
  • Jeong, J. H., Resop, J. P., Mueller, N. D., Fleisher, D. H., Yun, K., Butler, E. E., et al. Random forests for global and regional crop yield predictions. PLoS One 2016, 11(6), https://doi.org/10.1371/journal.pone.0156571
  • Lamorski, K., Šimůnek, J., Sławiński, C., Lamorska, J. An estimation of the main wetting branch of the soil water retention curve based on its main drying branch using the machine learning method. Water Resources Research 2017, 53, 1539–1552. https://doi.org/10.1002/ 2016WR019533
  • Povak, N. A., Hessburg, P. F., McDonnell, T. C., Reynolds, K. M., Sullivan, T. J., Salter, R. B., Cosby, B. J. Machine learning and linear regression models to predict catchment-level base cation weathering rates across the southern Appalachian Mountain region, USA. Water Resources Research 2014, 50, 2798–2814. https://doi.org/10.1002/2013WR014203

Use of machine learning for environmental risk prediction:

  • Mosavi, A., Ozturk, P., & Chau, K. W. (2018). Flood prediction using machine learning models: Literature review. Water, 10(11), 1536.
  • Choubin, B., Mosavi, A., Alamdarloo, E. H., Hosseini, F. S., Shamshirband, S., Dashtekian, K., & Ghamisi, P. (2019). Earth fissure hazard prediction using machine learning models. Environmental research, 179, 108770.

Use of machine learning for Network traffic flow

  • Zander, S., Nguyen, T., & Armitage, G. (2005, November). Automated traffic classification and application identification using machine learning. In The IEEE Conference on Local Computer Networks 30th Anniversary (LCN'05) l (pp. 250-257). IEEE.) 
  • Yu, H., Wu, Z., Wang, S., Wang, Y., & Ma, X. (2017). Spatiotemporal recurrent convolutional networks for traffic prediction in transportation networks. Sensors, 17(7), 1501.
  • Nguyen, H., Kieu, L. M., Wen, T., & Cai, C. (2018). Deep learning methods in transportation domain: a review. IET Intelligent Transport Systems, 12(9), 998-1004.

Use of machine learning in geoscience

  • Tahmasebi, P., Kamrava, S., Bai, T., & Sahimi, M. (2020). Machine learning in geo-and environmental sciences: From small to large scale. Advances in Water Resources, 142, 103619.;
  • Sun, A. Y., & Scanlon, B. R. (2019). How can Big Data and machine learning benefit environment and water management: a survey of methods, applications, and future directions. Environmental Research Letters, 14(7), 073001.)
  • Lary, D. J., Alavi, A. H., Gandomi, A. H., & Walker, A. L. (2016). Machine learning in geosciences and remote sensing. Geoscience Frontiers, 7(1), 3-10.

Machine learning for Civil engineering problems

  • Reich, Y., & Barai, S. V. (1999). Evaluating machine learning models for engineering problems. Artificial Intelligence in Engineering, 13(3), 257-272.
  • Reich, Y. (1997). Machine learning techniques for civil engineering problems. Computer‐Aided Civil and Infrastructure Engineering, 12(4), 295-310.
  • Vadyala, S. R., Betgeri, S. N., Matthews, J. C., & Matthews, E. (2021). A review of physics-based machine learning in civil engineering. Results in Engineering, 100316.

Bioinformatics, biomedicine, Biochemical engineering

  • Vellido, A., Martín-Guerrero, J. D., & Lisboa, P. J. (2012, April). Making machine learning models interpretable. In ESANN (Vol. 12, pp. 163-17
  • Abraham, A., Pedregosa, F., Eickenberg, M., Gervais, P., Mueller, A., Kossaifi, J.,Gramfort A., Thirion, B., Varoquaux, G. (2014). Machine learning for neuroimaging with scikit-learn. Frontiers in neuroinformatics, 14.).
  • Park, C., Took, C. C., & Seong, J. K. (2018). Machine learning in biomedical engineering. Biomedical Engineering Letters, 8(1), 1-3.

Medical risk prediction

  • Goldstein, B. A., Navar, A. M., & Carter, R. E. (2017). Moving beyond regression techniques in cardiovascular risk prediction: applying machine learning to address analytic challenges. European heart journal, 38(23), 1805-1814.

#2 QUESTION Second, Section 4 is divided into 11 sub-sections. So many sub-sections make this section difficult to understand.

#2 ANSWER: According with this comment, I split the section 4 into section 4 and 5. Section 4 (General results) contains former subsections 4.1-4.7. I removed the number header of subsections 4.1-4.7 (which describe the generic technical peculiarities of machine learning models in the reviewed papers) to make the reading clearer, while maintaining still separate paragraphs. Moreover, I added the section 5  (Specific results), which collects the former subsections 4.8-4.11.This section describes the properties of the machine learning techniques used in the reviewed papers, the comparison between ML models, the results of testing hybrid or ensemble models and the  results of ML models used to reduce or correct errors in numerical models. I hope the reviewer will find this section now easier to read.

#3 QUESTION Section 4 should be improved to show the main objective.

#3 ANSWER: dear reviewer, in order to meet your comment, and according to my answer to comment #2, section 4 was split into sections 4 and 5. This makes in my opinion clearer the explanation of how to select a  machine learning technique based on either their general features (section 4) or the advantages and disadvantages arising from techniques comparison (section 5). In order to enhance the focus on the main objective (already mentioned in section 1), I added an introduction to section 5 ( LINES 1276-1281), describing the objectives of this study:

This section aims to furnish specific information to orient modellers choosing the appropriate machine learning approach based either on the properties of each of the examined model (e.g., the most used algorithms, model structure, tuning parameters: Section 5.1) or on advantages and disadvantages arising from the comparison between dif-ferent machine learning techniques (Sections 5.2, 5.3 and 5.4).

It can be that the results are not clearly referred to the previous review section, so I added some reference to the sections

  • Section 5.3: I added in yellow the reference to the previous descriptive section “The use of hybrid models and a combination of techniques for data pre-processing (described in section 3.3) allows significant improvement in each modelling phase.” See line 1439
  • Section 5.4: I added in yellow the reference to the previous descriptive section “This section summarizes the main features of the machine learning models used for error correction and reduction (described in section 3.4).” See lines 1463

I hope the reviewer will find these modifications effective.

#4 QUESTION: Section 2: the characteristics of each numerical model and machine learning model should be discussed, such as the merit and demerit.

#4 ANSWER: Dear reviewer, section 2 describes the techniques (both numerical and machine learning) used in the reviewed papers. The list of merit and demerit of numerical and machine learning models is, in fact, a result of the review, and is widely discussed in sections 5 and 6.

#5 QUESTION: Figures and tables: The quality of the figures and tables should be improved, especially Table 1

#5 ANSWER: I agree with this comment; however, I think that the quality of figures is low only due to the pdf printing of the manuscript, because I embedded high quality figures in the text.

Table 1, in particular, is very complex and long. I will ask to the editor which is the best option to make it clearer: can I split it into 2 tables? Or can I enlarge the margin of the page? I will also ask them about figures.

#6 QUESTION: There are so many abbreviations made this paper difficult to read

#6 ANSWER: According with this comment and with comment #4 of reviewer 2, I removed most of abbreviations: PB (replaced with physically-based); ML (replaced with machine learning); MLP (replaced with multi-layer perceptron); DDM (replaced with data-driven models). I kept abbreviations for groundwater level (GWL), because in my opinion helps to speed the reading of the text. As for the abbreviations for machine learning models, I kept the abbreviation but in many cases such as in section 5, I added also the extended form.

#7 QUESTION The author concluded the numerical models and ML techniques can be successfully used as complementary to each other as a powerful groundwater management tool. But the reason for the conclusion is not sufficient. Firstly, the author only gave 16 reviewed papers. If the integrated method of numerical model and ML techniques is better, why there are so few studies on the integrated method. The author should give more explanation.

#7 ANSWER: Dear reviewer, according to your suggestion I added in lines 1545-1572 an explanation of why, in my opinion, there are so few studies on the integrated method:

“A lot of studies exist, concerning the use of machine learning models for groundwater simulation, developed on the basis of a limited number of observation points, without comparing results with numerical models. Conversely, the comparison of numerical and machine learning models is still a scarcely diffused task. This can be due to the fact that there are currently no well-defined procedures for the use of Machine learning techniques to enhance results of numerical models. Each modeler uses the machine learning tech-niques for fixing a specific weakness of the numerical model, of to ameliorate poor fitting between simulated and observed values; in most cases, modellers explore different ma-chine learning techniques to establish which one adapts better to its scopes. Another rea-son can be that the modeler should be familiar with both numerical and data driven mod-els, to correctly use both model types. Indeed, even if machine learning modelling doesn’t consider the behaviour of the natural system, a certain degree of knowledge about the hy-drological parameters and how they affect the results is required, in order to avoid, for example, model overfitting (which means fitting the model to all the input parameters, preventing the generalization ability of model- which is, on turn, given from the parame-ters effectively influencing groundwater level). In other words, the modeler should be able to manage both physically based data and statistical distributions of data, coupling dif-ferent skills: those typical of hydrogeologist and those typical of statistic/mathematics. In many cases, a modeler (or a team of modellers) can meet both these requirements, but it is not so common. In addition, machine learning models are viewed with some skepticism by numerical modellers. Physically based represent the technique most widely diffused and used by local administrators for groundwater management. Usually, the results of a physically based model are improved by the integrating new observations (when availa-ble) or by tuning model parameters in order to modify the conceptual model. The machine learning approach, instead, aims at detecting the inherent mechanism, increasing predic-tion skills without deriving this from physical knowledge. This ‘black box’ nature, where no insight is gained into how the model generated the solution, is not widely accepted to numerical modellers and can prevent the use of machine learning models.”

#8 QUESTION: Furthermore, the comparison of the simulated result, such as groundwater level, between the integrated method and the simple numerical model is benefit to support the conclusion.

#8 ANSWER: With respect to this comment, in my opinion it have been extensively mentioned how machine learning models outperform numerical models in the simulation of groundwater level, especially in section 3. It is also mentioned how the performance of models was calculated, i.e. the statistical indicators which were used in each case Some example below:

Lines 761-762, comparison between machine learning models ANN/ANFIS and numerical model

Lines 793-794 comparison between models ANN, RBF, SVM and numerical model

This is reiterate also in the discussion section (see for example lines 1629-163; 1683-1684)

Reviewer 2 Report

Summery

 The paper entitled “Improving results of existing groundwater numerical models using machine learning techniques: a review” by Cristian Di Salvo, reviews on groundwater level modeling techniques based on both numerical groundwater modeling (a physical and mathematical representation of hydrogeological regime) and data driven algorithms. Specifically, how to use optimally each method based on the data we have, for example model discretization (model complexity) and computational demands.  

The author discussed each advantage and disadvantage over each other, i.e. numerical model and machine learning.  The author highlights that machine algorithm can be useful to at local scale, such at observation well water level fluctuations. While it lacks the proper representation of the physical region of groundwater fluxes, where the numerical model performs better.

One concern I have is the author was diplomatic enough to describe that both are useful and complementary each other, however, how are complementary to each other? Are they used on one model run (both the numerical modeling and machine learning algorithm) or independently but support each other? Section 1, 2 and 3 are informative, well written and well written.

 In addition to one concern, I have two major comments. 1) how many previous literatures is used to conduct the literature from physical based, machine learning and couple physical and machine learning methods in groundwater studies? Explicitly. 2) are you considering water quantities (such as water level fluctuation of other fluxes or groundwater quality in your review) why not the groundwater quality?

Minor comments are:

Please can you remove abbreviations as much as possible, it is very hard to memorize all those abbreviations for your readers. For example: PB (line 26); ML (line 8); DDM (line 9) etc.

Line 54-55 please add reference from broader perspective (is it from hydrological, groundwater, medical or general sciences)

Line 93-135 on Section 2.1 Physical models such as, MODFLOW, Princeton Transport Code, SHETRAN are discussed. Can you also have a paragraphs for more groundwater modeling tools GSFLOW (USGS website), ParFlow (https://parflow.org/) , may be PHREEQC?

Line 302-308. In support of my major comment (1) can you divided the number of (16) paper into the comment I mentioned earlier. Thanks.

Line 325: Table1. Excellent work for the table and its content. Can you take it to consideration the geology or Hydrostratigraphic nature of the aquifer or the groundwater basin as one column? It might help which aquifer system is well presented in the physical based and data driven algorithms. (May be also model uncertainty if it exists)

Line 782: Nice data presentation for Figure 9. Does this represent model calibration, model optimization or parameter estimation? If so, what should be the percentage of the training in order to get better test result and validation based on your current studies.

Author Response

 Dear reviewer

please find here my answer to your comment. I am grateful to you for giving me the opportunity to improve the manuscript.

Comments and Suggestions for Authors

#1 QUESTION One concern I have is the author was diplomatic enough to describe that both are useful and complementary each other, however, how are complementary to each other? Are they used on one model run (both the numerical modeling and machine learning algorithm) or independently but support each other? Section 1, 2 and 3 are informative, well written and well written.

#1 ANSWER: Dear reviewer, I used the term complementary in the meaning of “run in parallel”, which means that each model is run independently, but uses the results of the other model. Specifically, in the majority of cases the results of physically based models previously run are used as dataset to train and test machine learning models and compare results. In [79] and [85], the machine learning models are trained using numerical model errors instead of model results; also in this case models are run independently. In Michael et al. [82], after training machine learning models with the results of a modflow model, new dataset derived from additional modflow run are used to update the machine learning models; in this case, the models are then run independently but there is a sequential (‘hierarchical’) series of runs.

According to this comment, I added a sentence to make this statement clearer (LINES 569-584)

#2 QUESTION:  In addition to one concern, I have two major comments. 1) how many previous literatures is used to conduct the literature from physical based, machine learning and couple physical and machine learning methods in groundwater studies? Explicitly.

#2 ANSWER: Dear reviewer, I can’t fully understand this comment, so I hope my answer will be appropriate. All the 16 papers reviewed in my manuscript concern the use of both machine learning and physically based models. I didn’t consider papers which treat the use of only one type of model (either machine learning or physically based). In fact, even if there is a lot of literature treating the use of machine learning for hydrological and groundwater problems (which are cited in the introduction), the focus of my paper is only on those who compare numerical and machine learning. Of course, I searched and red papers concerning the features and limitation of either machine learning techniques or physically based models, in order to adequately describe those in the manuscript. These are part of the references cited in chapters 1 and 2.

#3 QUESTION: are you considering water quantities (such as water level fluctuation of other fluxes or groundwater quality in your review) why not the groundwater quality?

#3 ANSWER: Dear reviewer, It is not clear to me if you mean strictly modelling groundwater quality or you mean contaminant transport. First of all, I choose to focus the review only on groundwater quantity mainly because this is my main work field and I am not familiar with modelling water quality. If I’m not wrong, water quality models (for example: Water Quality Analysis Simulation Program by EPA) are not necessarily linked to a hydrodynamic (i.e., physically based) model, so they don’t account for the spatial distribution of aquifer’s parameters. Clearly, this family of quality models differs very much from the numerical-physically based models which simulate groundwater flow, and including these in the review would be misleading. On the other hand, your comment opens an important question about the transport models (used to model the movement of pollutant in groundwater). I am thinking for example to MT3D by USGS. It would be surely interesting to compare the results of solute transport numerical and machine learning models. However, modelling groundwater solute transport with numerical models implies setting up a robust “quantity “ model. That is, you need first to check whether your model is capable to reproduce groundwater fluxes or oscillations, before examining, for example, the plume of a pollutant, of the contaminant advection/dispersion. For this reason, in my opinion, a modeller should first focus on searching for the best modelling technique for groundwater flow, and then start working on solute transport. The subject of the review paper is exactly focused on this first phase. However, following your interesting suggestion, I added a sentence in the discussion, in which I state that a comparison between physically based and ML models focused on groundwater solute transport should be the subject of further research. ( lines 1704-1708)

Minor comments are:

#4 QUESTION: Please can you remove abbreviations as much as possible, it is very hard to memorize all those abbreviations for your readers. For example: PB (line 26); ML (line 8); DDM (line 9) etc.

#4 ANSWER: Following your suggestion, I removed the abbreviations: PB (replaced with physically-based); ML (replaced with machine learning); MLP (replaced with multi-layer perceptron); DDM (replaced with data-driven models). I kept abbreviations for groundwater level (GWL), because in my opinion helps to speed the reading of the text. As for the abbreviations for machine learning models, I kept the abbreviation but in many cases such as in section 5, I added also the extended form.

#5 QUESTION: Line 54-55 please add reference from broader perspective (is it from hydrological, groundwater, medical or general sciences)

#5 ANSWER According to the suggestion, I added in the introduction many much more  cited literature (see answer to comment #1 of first reviewer)

#5 QUESTION: Line 93-135 on Section 2.1 Physical models such as, MODFLOW, Princeton Transport Code, SHETRAN are discussed. Can you also have a paragraphs for more groundwater modeling tools GSFLOW (USGS website), ParFlow (https://parflow.org/) , may be PHREEQC?

#5 ANSWER: Dear reviewer, I described only the physical models which are used in the reviewed papers, and adding physical based codes which are not part of the review would be in my opinion misleading. To note, I didn’t find any paper concerning the use of GSFLOW and machine learning models. PHREEQC is used for modelling water quality, which is not the subject of this review. Regarding ParFlow, following your comment, I searched and found this paper concerning the coupled use of ParFlow and machine learnig models: Tran, H., Leonarduzzi, E., De la Fuente, L., Hull, R. B., Bansal, V., Chennault, C., ... & Maxwell, R. M. (2021). Development of a deep learning emulator for a distributed groundwater–surface water model: Parflow-ml. Water13(23), 3393.). However, this last is about deep learning coupling within a physically based models, which is not specifically the focus of the paper; including this in the review paper would mean adding explanation about deep learning model, in addition to notably expanding the review and the paper length.

#6 QUESTION: Line 302-308. In support of my major comment (1*) can you divided the number of (16) paper into the comment I mentioned earlier. Thanks.

#6 ANSWER: Dear reviewer, as already answered to your comment #2, I can’t fully understand this comment and I hope my answer will be appropriate. All the 16 papers reviewed in my manuscript concern the use of both machine learning and physically based models. I didn’t consider papers which treat the use of only one type of model (either machine learning or physically based). In fact, even if there is a lot of literature treating the use of machine learning for hydrological and groundwater problems (I cited it in the introduction), the focus of my paper is only on those who compare numerical and machine learning. Of course, I searched and red papers concerning the features and limitation of either machine learning techniques or physically based models, in order to adequately describe those in the manuscript. These are the references cited in chapters 1 and 2.

In order to clarify this, I modified lines 302-308 (now lines 554-556):

“Throughout our research, few papers were found in literature that examine the use of both numerical models and machine learning models in GWL forecasting. Here….”

#7 QUESTION: Line 325: Table1. Excellent work for the table and its content. Can you take it to consideration the geology or Hydrostratigraphic nature of the aquifer or the groundwater basin as one column? It might help which aquifer system is well presented in the physical based and data driven algorithms. (May be also model uncertainty if it exists)

#7 ANSWER: Dear reviewer, as you suggested, I added a column with the hydrostratigraphy of the aquifer or groundwater basin to table. Due to the increased dimension of the table, I will propose to split it into 2 separate tables.

#8 QUESTION: Line 782: Nice data presentation for Figure 9. Does this represent model calibration, model optimization or parameter estimation?

#8 ANSWER: Dear reviewer, figure 9 shows the percentages of data used for (i) model training (the phase in which the machine learning model “learns” from the input data and produces a function representing the system behaviour. In this phase, the coefficient of the function-namely parameters- are estimated) and (ii) testing (where the performance of machine learning models is evaluated); some authors split data into three parts: for training, testing and validating the model, (validating means checking if the prediction function effectively works by using new input data to the model). It is explained in section 2.2.1. However, following your suggestion, I added a brief explanation of these elements also in section 4-Subset for ML model training, validation and testing (lines 1176-1181).

#9 QUESTION:  If so, what should be the percentage of the training in order to get better test result and validation based on your current studies.

#9 ANSWER: As concluded in section 4-Subset for ML model training, validation and testing, in the reviewed studies there is no mention about the best subdivision into training and testig dataset. However, it can be noted that in all cases (except Sahoo et al., 2017) the dataset for training in the reviewed papers was always at least the 60% (Figure 9), reaching the 95%. In the majority of the papers, (9 among 16), the percent of training dataset exceeds the 80%. With regard to testing dataset, authors use a percentage highly variable, between 4.5 and 40%. It can be concluded that a robust machine learning model should always be based on at least the 60% of training data, and 40% of testing data.

Following the comment, I modified this section adding lines (for example 1185-1186; 1194-1195)

Round 2

Reviewer 1 Report

The structure of the manusrcipt still needs to be improved. Abbreviations are defined repeatedly, for example, ANNs is repeatedly defined on page 92, 149, 857, and 860y.

Author Response

Dear Reviewer, during the first manuscript reviewing process, I deleted some of the abbreviations following your suggestions (e.g., PB in place of physically base; GWL in place of groundwater level). At the same time, I added the extended name for the abbreviations of machine learning models in some section, to recall the definition of the abbreviation. For this reason, there are some repeated definitions, as in the lines that you mentioned. It seems that this last choice didn’t improve efficiently the reading, so I changed again: for machine learning models, I removed abbreviations in section 1; I defined the abbreviation only once, in section 2.2, when all the machine learning models are described. In section 5.1, I removed definitions (see line 892); further, I removed abbreviations from the header in lines 893, 935, 940, 947, 960, 966. Finally, I removed the extend form “gradient boosted regression tree” in line 559 because the abbreviation is already defined in line 349. I hope these corrections will improve the reading of the paper.

Morover, I corrected some inaccuracies in the discussion section (lines1220-1226), with the attempt to ameliorate the text.

Finally, I found new interesting literature concerning the subject of this paper, and I added the following references:

Fienen, M. N., Nolan, B. T., Feinstein, D. T., Starn, J. J. Metamodels to bridge the gap between modeling and decision support. USGS Staff -- Published Research 2015. 860.

Fienen, M. N., Nolan, B. T., Kauffman, L. J., & Feinstein, D. T.  Metamodeling for groundwater age forecasting in the Lake Michigan Basin. Water Resources Research 201854(7), 4750-4766.

I removed a citation which was not pertinent:

Goldstein, B. A., Navar, A. M., & Carter, R. E. Moving beyond regression techniques in cardiovascular risk prediction: applying machine learning to address analytic challenges. European heart journal 2017, 38(23), 1805-1814.

Unfortunately, I didn't receive any answer about how to improve the resolution of table 1

Round 3

Reviewer 1 Report

Thanks for your efforts. Now it can be accepted

Author Response

Thank you for appreciating my effort. I am glad you consider the manuscript now acceptable for being published. I attach here the new file with some corrections requested by the editor. 

My best regards

Cristina
